# Exploiting lung adaptation and phage steering to clear pan-resistant Pseudomonas aeruginosa infections in vivo

Eleri A. Ashworth[1], Rosanna C. T. Wright [2], Rebecca K. Shears [1,4], Janet K. L. Wong[1], Akram Hassan[1], James P. J. Hall [3], Aras Kadioglu[1,5] ✉ & Joanne L. Fothergill [1,5] ✉

*Pseudomonas aeruginosa* is a major nosocomial pathogen that causes severe disease including sepsis. Carbapenem-resistant *P. aeruginosa* is recognised by the World Health Organisation as a priority 1 pathogen, with urgent need for new therapeutics. As such, there is renewed interest in using bacteriophages as a therapeutic. However, the dynamics of treating pan-resistant *P. aeruginosa* with phage in vivo are poorly understood. Using a pan-resistant *P. aeruginosa* in vivo infection model, phage therapy displays strong therapeutic potential, clearing infection from the blood, kidneys, and spleen. Remaining bacteria in the lungs and liver displays phage resistance due to limiting phage adsorption. Yet, resistance to phage results in re-sensitisation to a wide range of antibiotics. In this work, we use phage steering in vivo, pre-exposing a pan resistant *P. aeruginosa* infection with a phage cocktail to re-sensitise bacteria to antibiotics, clearing the infection from all organs.

Antibiotic resistance is a global health issue and the need for new therapeutics was highlighted by the World Health Organisation (WHO), with *Pseudomonas aeruginosa* deemed a priority 1 pathogen in urgent need of new therapeutic strategies[1]. *P. aeruginosa* occurs naturally in various environments and can cause nosocomial, life-threatening diseases including burn wound and pulmonary infections. It also causes acute issues in hospitals due to its ability to cause sepsis and its intrinsic resistance to many classes of antibiotics.

Hospital-acquired pneumonia (HAP) is the leading cause of death from nosocomial infections in critically ill patients. In 2017, the European Centre for Disease Prevention and Control (ECDC) reported that 8.3% of patients in ICU for >2 days presented with HAP and the most frequently isolated organism was *P. aeruginosa*, with carbapenem resistance reported in 26% of isolates[2,3]. Furthermore, *P. aeruginosa* is the third most prevalent Gram-negative species isolated from clinical bacteraemia[4,5], and *P. aeruginosa* associated bacteraemia is linked with increased mortality compared with bacteraemia caused by other Gram-negative pathogens[6]. This highlights the critical importance of developing effective new therapeutics against pan-resistant pathogens.

Since the advent of increasing antibiotic resistance, there is renewed interest in using bacteriophages to treat bacterial infections. Phage therapy has many advantages compared to antibiotics, such as increased specificity, replication at the site of infection, low manufacturing cost and little reported toxicity[7]. Phage therapy has also shown efficacy in a variety of in vivo murine models[8–11]. In a *P. aeruginosa* chronic lung infection murine model, phage PELP20 was highly effective against an established 6-day lung infection, completely clearing bacteria from the lungs in 70% of mice[12]. However, complex evolutionary phage-bacteria dynamics in vivo remain unclear as environmental factors present in vivo, such as limited oxygen availability, and the presence of mucin and polyamines, have been shown to contribute to phage resistance[13–15]. Thus far, bacterial adaptation to in vivo factors and the effect on phage resistance has not been explored.

[1]Department of Clinical Infection, Microbiology and Immunology, University of Liverpool, Liverpool, UK. [2]Division of Evolution & Genomic Sciences, School of Biological Sciences, University of Manchester, Manchester, UK. [3]Department of Evolution, Ecology and Behaviour, University of Liverpool, Liverpool, UK. [4]Present address: Centre for Bioscience, Manchester Metropolitan University, Manchester M1 5DG, UK. [5]These authors contributed equally: Aras Kadioglu, Joanne L. Fothergill. ✉e-mail: ak68@liverpool.ac.uk; j.fothergill@liv.ac.uk

Previous case studies show that phage therapy is effective clinically in the management of extensive pulmonary *P. aeruginosa* infections[16,17]. Additionally, phage therapy can "re-sensitise" previously antibiotic resistant bacteria to antibiotics in vitro, a concept known as phage steering[18–20]. However, to date, phage steering has not been demonstrated to clear an infection in vivo. Overall, bacteriophage therapy is an exciting potential alternative for treating multi-drug resistant (MDR) *P. aeruginosa* infections, needing further exploration.

In this study, we developed an in vivo model for *P. aeruginosa* systemic infection using a megaplasmid carrying, pan-resistant *P. aeruginosa* strain[21]. Our phage cocktail had strong therapeutic potential within this in vivo model. We found that phage resistance developed in vivo even the absence of phage treatment, with bacteria reisolated from the lungs of untreated mice displaying increased phage resistance. When investigated further, we found bacterial adaptation to in vivo factors such as oxygen availability, and the presence of mucin and polyamines contributes to the development of phage resistance. We also observed that bacterial isolates from phage treated mice gained phage resistance by limiting phage adsorption but were more susceptible to a wide range of antibiotics, resulting in a shift from carbapenem resistant to susceptible. The therapeutic potential for phage steering was demonstrated as pre-exposure to the phage cocktail re-sensitised the infection to antibiotics, permitting bacterial clearance in our pan-resistant *P. aeruginosa* in vivo infection model.

## Results

### Phage infectivity of clinical *P. aeruginosa* isolates

The susceptibility of 551 clinical bacterial isolates from a range of sources, including the UK and Thailand, to the four phages (PELP20, PNM, PT6 and 14/1) used in our phage cocktail was determined (Supplementary Data 1, Supplementary Table 1). PELP20 has previously been shown to be effective in vivo against *P. aeruginosa* Liverpool Epidemic strains[12], whilst phages PNM and 14/1 have previously been included in a well-defined cocktail available for use in clinical human trials[22]. Here, PELP20 displayed the broadest infection range (Fig. 1a; >76% infectivity of 551 clinical isolates) and 14/1 the narrowest (Fig. 1d; >41% infectivity). Overall, 454 (>80%) clinical isolates were susceptible to ≥1 phage and 190 (~35%) isolates demonstrated at least intermediate susceptibility to all 4 phages (Fig. 1f). All four phages were effective against the pan-resistant B9 (T2436) strain used in our mouse model (Fig. 1).

### Invasive respiratory model for pan-resistant *P. aeruginosa*

An invasive respiratory infection model of pan-resistant *P. aeruginosa* was developed to test efficacy of phage therapy in vivo. Four different strains of MDR *P. aeruginosa* isolated from clinical sputum samples from patients in Thailand were tested; these included strains B3 (T2101), B8 (T2584), B9 (T2436), and C7 (T3582) (Supplementary Fig. 1). Mice were intra-nasally infected with each strain at $10^6$ cfu/ml and $10^7$ cfu/ml challenge doses, and the bacterial load in the lungs, liver, blood, spleen and kidneys was determined over a 48 h period post-infection. Secondary to lung infection, systemic spread was observed with all four strains at both challenge doses. Strain B9 (T2436) was chosen to continue forward to phage testing, as it had consistently high CFU loads in all organs and was both pan-resistant and harboured an unusual megaplasmid carrying antimicrobial resistance (AMR) genes[21].

Following respiratory infection with B9, bacteria were found in the lungs (>$10^4$ cfu/ml) and liver (>$10^3$ cfu/ml) within 6 h post-infection (Supplementary Fig. 2a, b) while bacteraemia developed rapidly by 4 h in blood (>$10^4$ cfu/ml) (Supplementary Fig. 2c). By 24 h post-infection, bacteria could also be isolated from the kidneys and spleen (Supplementary Fig. 2d, e). The bacterial load in each of these infection sites were maintained for the 48 h infection period, with CFU load increasing in liver, kidney, and spleen.

### Early phage treatment significantly reduces *P. aeruginosa* in vivo

To investigate the timing and route of administration, treatment was trialled both early (simultaneously with bacteria) and late (5 h following bacteraemia) with administration via the intranasal and intravenous routes.

In mice treated with phage immediately after bacterial infection via the intranasal route (Fig. 2), phage treatment was able to target the site of infection (the lungs) before bacteria had entered the bloodstream. A single phage treatment of either the 4-phage cocktail or PELP20 alone were trialled. Compared to mock treated mice, phage cocktail treated and PELP20 treated mice had significantly reduced bacterial load in the lung ($p < 0.0001$), liver ($p = 0.0013$), and blood ($p = 0.0021$) 24 h post-infection, and significantly reduced bacterial load in the kidneys ($p = 0.002$) and spleen ($p = 0.0001$) 48 h post-infection (Fig. 2).The phage cocktail was more effective than PELP20 alone: phage cocktail treated mice completely cleared the infection from all organs tested, except from the liver which had residual CFU left ($\leq 10^2$ cfu/ml). In comparison, bacteria were still present in the lungs (<$10^4$ cfu/ml), liver (<$10^2$ cfu/ml), kidney (<$10^2$ cfu/ml) and spleen (<$10^1$ cfu/ml) of PELP20 treated mice, 48 h post-infection (Fig. 2).

To determine whether intranasal administration of the phage cocktail resulted in phage dissemination into the bloodstream and amplification in vivo (i.e., auto-dosing), phage density (plaque forming units; PFU) was determined at 48 h for both uninfected and B9 infected mice treated with the phage cocktail. Intranasal administration of phage cocktail reached all organs tested, and in most animals, persisted for 48 h even in the absence of bacterial infection (Fig. 2f). Significant phage amplification was seen in the lungs of B9 infected mice, as the PFU was significantly increased ($p = 0.0051$) compared with uninfected, phage cocktail treated mice. In summary, early respiratory phage treatment (prior to systemic spread of *P. aeruginosa*), significantly reduced or totally cleared bacterial infection from all tissue sites.

### Delayed phage treatment significantly reduces *P. aeruginosa* in vivo

We next determined whether phage treatment administered after systemic spread of *P. aeruginosa*, could reduce bacterial load. The efficacy of both intranasal and intravenous delivery of phage was tested. To mimic treating a MDR systemic infection, phage treatment was administered 5 h post-infection, after infection had become systemic. Intranasal administration of PELP20 5 h post infection was not effective at reducing CFU loads, however, treatment with phage cocktail significantly reduced (or cleared) CFU loads in lung, liver and blood by 24–48 h post-infection, compared to mock treated mice (Supplementary Fig. 3).

Intravenous treatment with phage was significantly more efficacious than intranasal administration, with significantly reduced bacterial loads in the lung ($p = 0.0021$), and blood ($p = 0.014$) 24 h post infection; and significantly reduced bacterial loads in the liver ($p = 0.0007$), kidneys ($p = 0.002$) and spleen ($p = 0.0031$) 48 h post-infection compared to mock-treated mice. Although bacteria were still detectable at low levels in the lungs ($\leq 10^2$ cfu/ml) and liver ($\leq 10^2$ cfu/ml) in phage cocktail treated mice, no bacteria were detected in the blood, kidneys and spleen at 48 h post infection (Fig. 3). In contrast, in single phage PELP20 treated mice, at 48 h the infection was only cleared from blood. Similar to intranasal delivery, when PFU was measured 48 h after intravenous administration of the phage cocktail, phages could be detected in all tissues tested, including significant phage amplification in the lungs of B9 infected mice ($p < 0.0001$) (Fig. 3f). The phage cocktail was therefore highly effective at reducing (and in some cases even clearing) CFU from infected tissues and blood in this challenging systemic infection model.

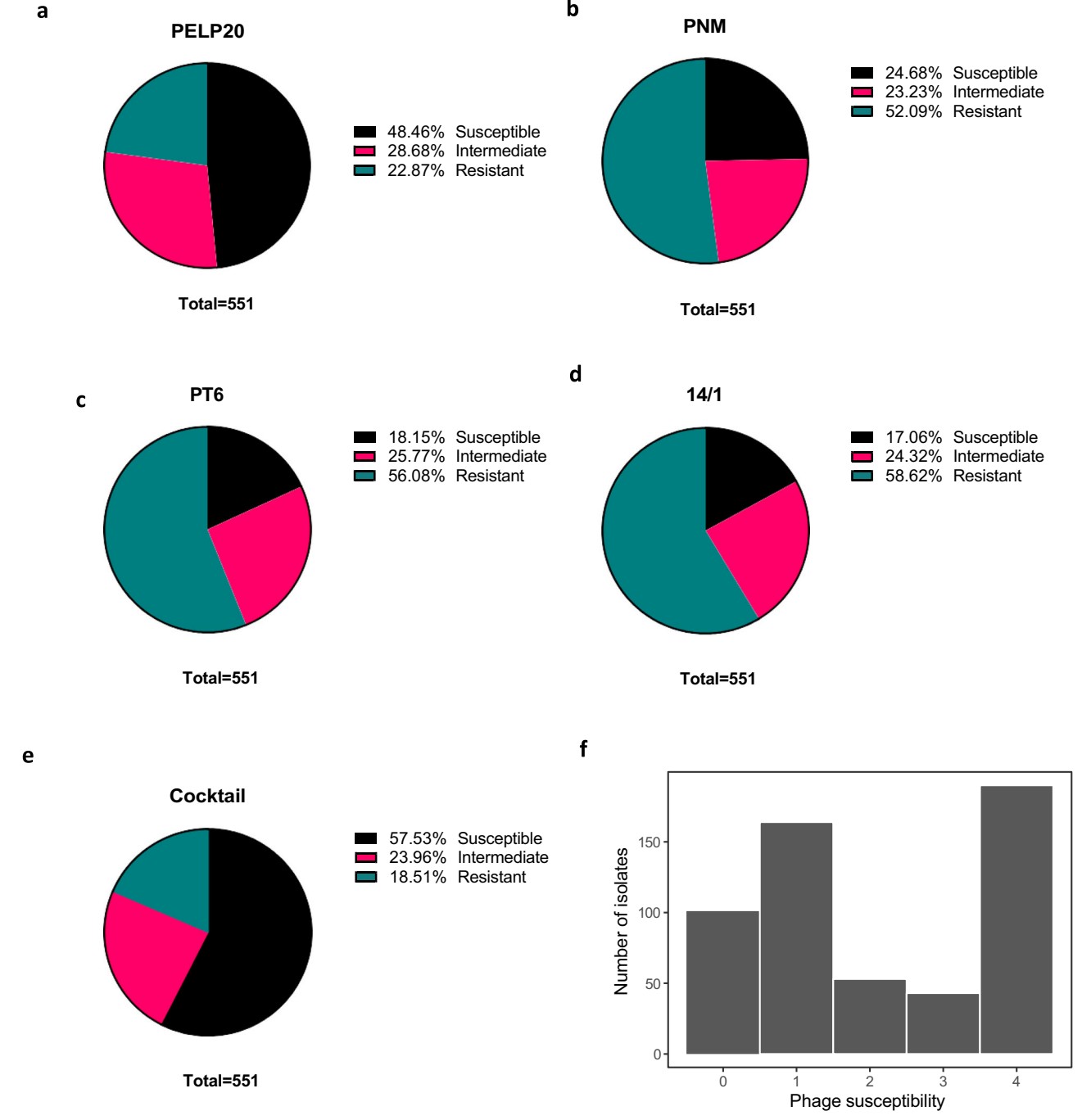

**Fig. 1 | Phage cocktail screening against 551 clinical isolates.** Percentage of isolates susceptible (black - complete lysis), intermediate (pink - incomplete lysis) and resistant (green - no lysis) determined by direct spot test method for phages **a**) PELP20 **b**) PNM **c**) PT6 **d**) 14/1 **e**) cocktail and **f**) Frequency of phage susceptibility across clinical isolates measured as number of phages each isolate shows susceptible to (including Susceptible and Intermediate). Source data are provided as a Source Data file and Supplementary data file 1.

## *P. aeruginosa* develops resistance to phage in vivo

To determine whether resistance to phage developed in vivo, *P. aeruginosa* present in tissues at 48 h post-infection from mock-treated, phage cocktail treated and PELP20 treated mice were harvested. Efficiency of plating (EOP) of the phages present in the cocktail (PELP20, PNM, 14/1 and PT6) was conducted on all the in vivo recovered isolates to determine whether phage resistance developed in vivo.

Surprisingly, bacteria isolated from the lungs of mock-treated mice displayed increased phage resistance compared to the input bacterial strain (Fig. 4a, b). These non-phage treated isolates from the lungs were found to be nearly completely resistant to all the phages present in the cocktail (Fig. 4a, c). This has important implications, showing that resistance developed in the lungs independently of phage treatment, potentially limiting the efficacy of phage therapy in this niche. However, the development of resistance in non-phage treated isolates was not universal across all organs. Isolates from the kidneys were still susceptible to the input phages, while phage resistance in isolates recovered from the liver, blood and spleen was more variable (Fig. 4a). This indicates that some biological niches may be more difficult to treat using phage therapy due to resistance developing independently of phage treatment.

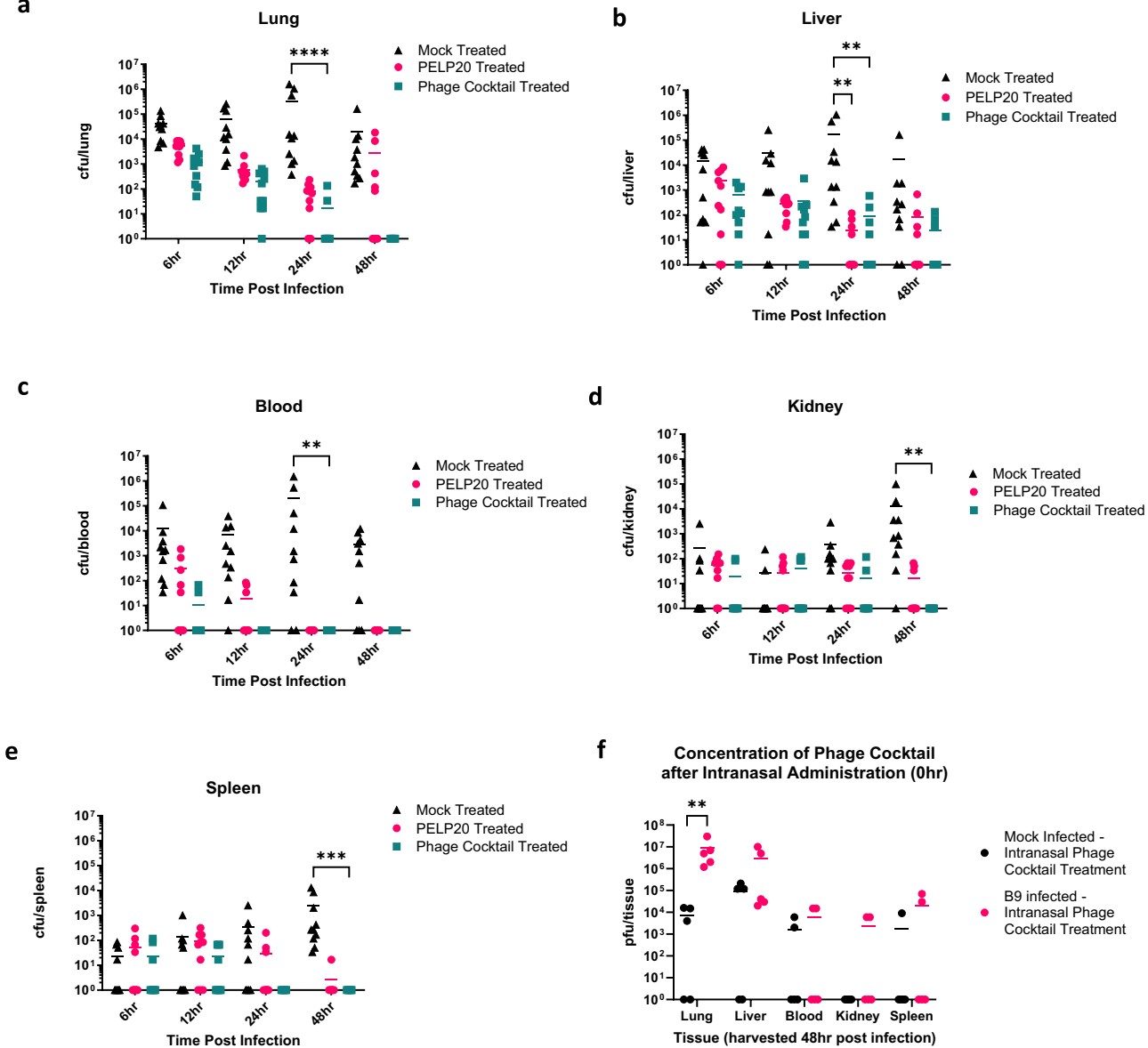

**Fig. 2 | Early intranasal phage treatment in vivo.** Bacterial colony forming units (CFU) following intranasal treatment with either phosphate buffered saline (PBS) (black triangles), PELP20 (pink circles), or phage cocktail (green squares) immediately after *P. aeruginosa* infection. **a**–**e** CFU measured in the lungs, liver, blood, kidneys, and spleen. Each symbol represents an individual mouse and line indicates the mean. Results are combination of two independent experiments, n = 10 mice per group per timepoint. Significant difference in CFU compared to mock treated mice was observed in the lungs (*p* < 0.0001, phage cocktail treated mice), liver (*p* = 0.0013, for both PELP20 and phage cocktail treated mice) and blood (*p* = 0.0021, phage cocktail treated mice) at 24 h and kidney (*p* = 0.0044, phage cocktail treated mice) and spleen (*p* = 0.0001, phage cocktail treated mice) at 48 h.

**f** Shows phage plaque forming units (PFU) in the tissues of naïve mice treated with phage cocktail (black circles) and *P. aeruginosa* infected mice treated with intranasal phage cocktail immediately following bacterial infection (pink circles). Each symbol represents an individual mouse and line indicates mean. Results are combination of 1 independent experiment, n = 5 mice per group. Significant differences in PFU between naïve mice and *P. aeruginosa* infected mice treated with phage cocktail in the lungs was observed, *p* = 0.0051. The y-axis has been corrected by adding 1 (to zero all samples). Statistics were performed using a two-way ANOVA Bonferroni correction ****p* < 0.0001 ****p* < 0.001 ***p* < 0.01 **p* < 0.05. Source data are provided as a Source Data file.

Organ-related differences in resistance was observed when comparing the non-phage treated isolates and phage treated isolates (Fig. 4b & Supplementary Fig. 4): whilst phage treated isolates from the lungs were slightly less resistant to the input phages than mock-treated isolates, phage treated isolates from the liver and kidneys developed more resistance compared to the mock-treated isolates. This suggests that bacterial adaption to environmental factors could impact the development of phage resistance (Fig. 4b). We then explored whether phage resistance was due to modification of phage surface receptors via phage adsorption assays.

Non-phage treated bacterial isolates had variable phage adsorption rates, with only isolates from the lungs showing no adsorption, explaining the phage resistance observed for non-phage treated lung isolates. All phage treated isolates were resistant to the input phages due to reduced adsorption (Fig. 4c, d). This may indicate that phage treated bacteria either modify phage receptors or alter their outer membrane properties resulting in phage resistance. Overall, while our phage cocktail showed efficacy, phage resistance still occurs in vivo, which has important implications for future use of phage therapy.

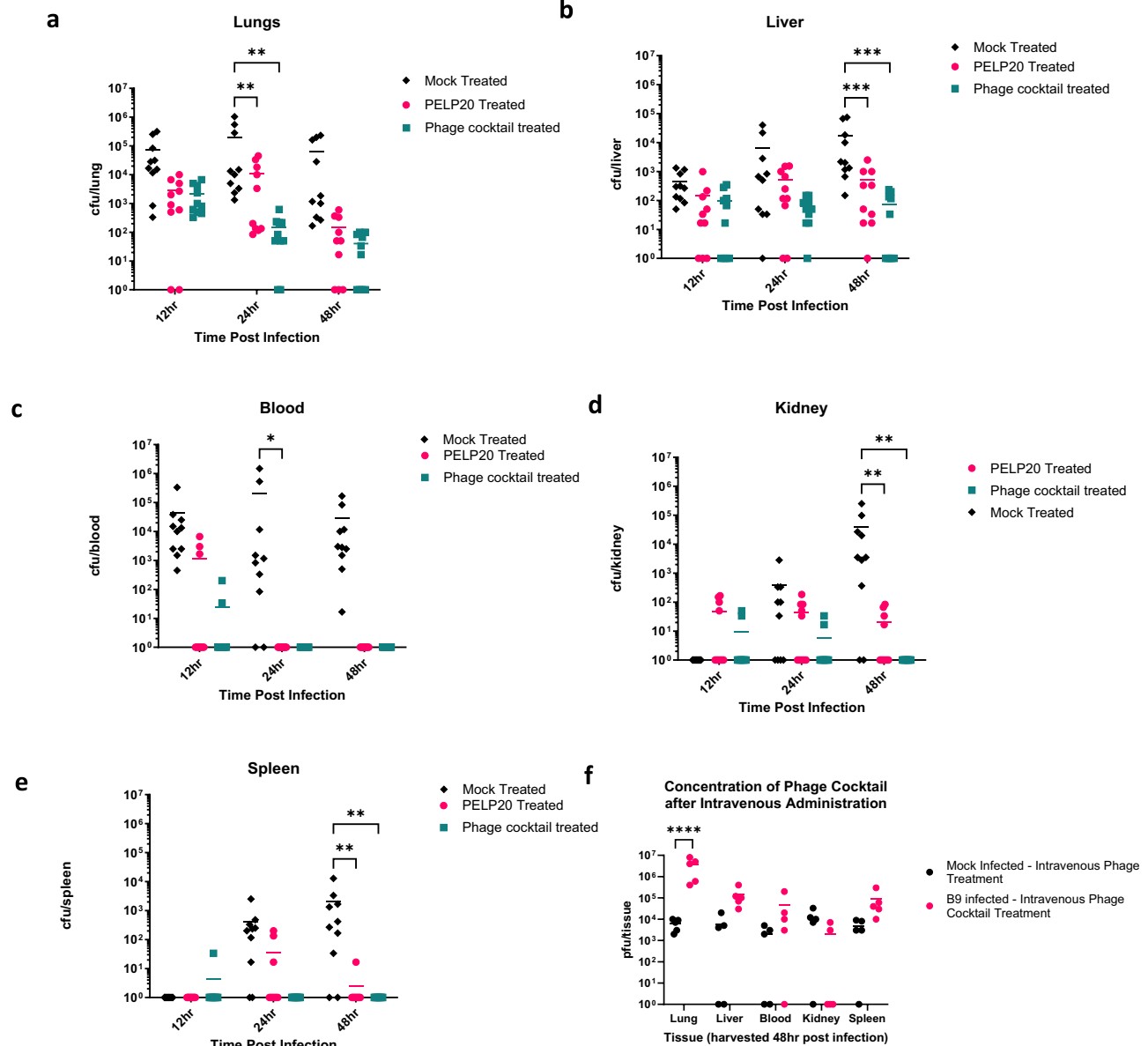

**Fig. 3 | Delayed intravenous phage treatment in vivo.** Bacterial colony forming units (CFU) following intravenous treatment with either phosphate buffered saline (PBS) (black triangles), PELP20 (pink circles), or phage cocktail (green squares) 5 h after *P. aeruginosa* infection. **a–e** CFU measured in the lungs, liver, blood, kidneys, and spleen. Each symbol represents an individual mouse and line indicates the mean. Results are combination of two independent experiments, $n = 10$ mice per group per timepoint. Significant difference in CFU compared to mock treated mice was observed in the lungs ($p = 0.0038$ and $p = 0.0021$, for PELP20 and phage cocktail treated mice, respectively) and blood at 24 h ($p = 0.0142$, for PELP20 treated mice) and liver ($p = 0.0009$ and $p = 0.0007$, for PELP20 and phage cocktail treated mice, respectively), kidney ($p = 0.002$ for both PELP20 and phage cocktail

treated mice) and spleen ($p = 0.0025$ for both PELP20 and phage cocktail treated mice). **f** Shows phage plaque forming units (PFU) in the tissues of naïve mice treated with phage cocktail (black circles) and *P. aeruginosa* infected mice treated with intravenous phage cocktail 5 h post bacterial infection (pink circles. Each symbol represents an individual mouse and line indicates mean. Results are combination of 1 independent experiment, $n = 5$ mice per group. Significant differences in PFU between naïve mice and *P. aeruginosa* infected mice treated with phage cocktail in the lungs was observed, $p < 0.0001$. The y-axis has been corrected by adding 1 (to zero all samples). Statistics were performed using a two-way ANOVA Bonferroni correction ****$p < 0.0001$, ***$p < 0.001$, **$p < 0.01$, *$p < 0.05$. Source data are provided as a Source Data file.

### *P. aeruginosa* adaptation to the lung results in phage resistance

The lung environment is unique[23], with many environmental stressors to which *P. aeruginosa* adapts, which could result in phage resistance as a secondary effect. These factors include variable oxygen availability[13], the presence of polyamines[14] and mucin[15], all of which have been linked with phage resistance. To explore this, an experimental evolution approach was conducted, where B9 was grown in nutrient rich broth (LB Broth) or Healthy Lung Media (HLM)[23] for 48 h, then isolates were recovered and tested for phage resistance (via EOP). The effect of oxygen availability was determined by repeating

the experiment under microaerophilic and anaerobic conditions. Finally, the effect of mucin and polyamines was investigated by growing B9 in LB broth supplemented with either polyamines (spermidine-200 ng/ml, spermine-32.5ug/l, and putrescine-616 ug/l) or mucin (1.2 mg/ml), at the same concentrations used in HLM and testing recovered isolates for resistance via EOP compared to the input strain and controls (i.e., LB in aerobic conditions).

We found *P. aeruginosa* grown in media mimicking the lung environment acquired resistance to all four phages present in the cocktail (Fig. 5a). Furthermore, we show that even in nutrient rich

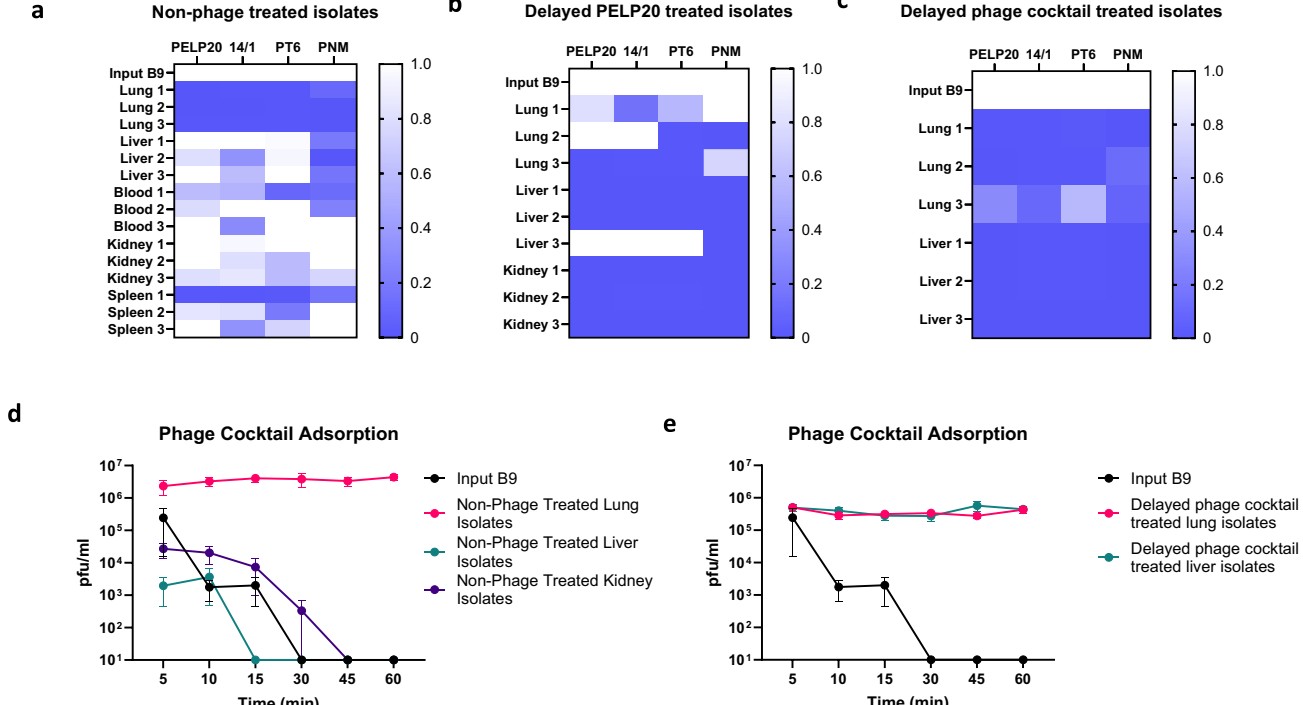

**Fig. 4 | Phage resistance develops in vivo due to reduced phage adsorption.**
**a** Heat map displaying phage resistance via efficiency of plating (EOP) of non-phage treated in vivo adapted isolates to the phages present in the cocktail. Each score represents susceptibility (white squares, score = 1) to resistance (dark blue squares, score = 0), $n = 3$ for each group. **b** Heat map illustrating phage resistance (via EOP) of isolates recovered from delayed PELP20 treated mice to the phages in the cocktail. Scores range from complete susceptibility (white squares, score = 1) to complete resistance (dark blue squares, score = 0), $n = 3$ for each group. **c** Heat map showing phage resistance (via efficiency of plating) of isolates recovered from delayed phage cocktail treated mice to the phages present in the cocktail. Scores range from complete susceptibility (white squares, score = 1) to complete resis-tance (dark blue squares, score = 0), $n = 3$ for each group. **d** Phage cocktail adsorption over time for the input isolate (black circles) and non-phage treated in vivo adapted isolates from the lungs (pink circles), liver (green circles), and kidney (purple circles). Mean with SEM is indicated, $n = 3$ for each group. **e** Phage cocktail adsorption over time for the input isolate (black circles) and phage cocktail treated in vivo adapted isolates from the lungs (pink circles) and liver (green cir-cles). Mean with SEM is indicated, $n = 3$ for each group. Source data are provided as a Source Data file.

broth, oxygen availability was a key driver of phage resistance (Fig. 5a). While bacteria grown in the presence of polyamines only developed resistance to the phage PNM, *P. aeruginosa* isolates grown in the pre-sence of mucin demonstrated resistance to all four phages (Fig. 5b).

We have shown that bacterial adaption to the lung, which includes limited oxygen availability, mucins, and polyamines, can result in phage resistance as a secondary effect. Bacterial adaption to different biological niches may reduce the efficacy of phage therapy and should be taken into consideration.

## Phage exposure alters antimicrobial susceptibility of *P. aeruginosa*

Previous in vitro studies showed that phage resistant bacteria can become re-sensitised to antibiotics[18–20]. However, there is little evi-dence for re-sensitisation within in vivo models. Therefore, altered antimicrobial sensitivity in vivo was explored via disk diffusion assays for a range of antibiotics and E-Test to determine the minimum inhi-bitory concentration (MIC) of two clinically relevant antibiotics, tobramycin and meropenem.

Non-phage treated bacteria displayed only small differences in antibiotic susceptibility compared to the input isolate (Fig. 6). For the majority of antibiotics, there was either no change or changes <5 mm in zone diameter and MIC for tobramycin and meropenem (Fig. 6a). For non-phage treated isolates from the lungs, the zone of inhibition to meropenem increased by 5 mm, however isolates were still classed as resistant overall. For bacteria isolated from other organs of non-phage treated mice, there was an increase in the zone of inhibition

to cefepime of between 5 and 9 mm (Fig. 6b–f) however, this did not result in a change in resistance based on EUCAST clinical breakpoints. Small increases in resistance to ticarcillin-clavulanic acid was observed and, in some cases, resulted in a change in resistance class (from intermediate to resistant) (Fig. 6f). While isolates derived from differ-ent in vivo niches displayed small differences in antibiotic suscept-ibility, the isolates from mock-treated mice were largely unchanged based on clinical breakpoints which are designed to indicate clinical efficacy in systemic infection.

However, isolates from delayed phage treatment group did demonstrate altered antibiotic susceptibility based on clinical break-points. When phage treatment of infected mice was delayed by 5 h, an increase in antibiotic susceptibility in phage resistant isolates was seen compared to the input strain. Large changes in antibiotic susceptibility were observed across multiple classes of antibiotics, typically a 15–25 mm increase in the zone of inhibition (Fig. 7a–e). Additionally, >2-fold reduction in MIC was seen in 14 out of 15 isolates for mer-openem and >4-fold reduction was in 11 out of 15 isolates for tobra-mycin (Fig. 7f). Across all isolates, only resistance to aztreonam remained unaltered when compared to the input isolate.

Of the 15 isolates from delayed phage treated mice, 13 isolates showed multiple shifts in susceptibility classification (Fig. 7). Isolates from delayed phage treated lungs and liver swung from meropenem-resistant to sensitive and for one isolate treated with the cocktail, there were 12 shifts in susceptibility classification. These results highlight major and clinically relevant, changes in antibiotic susceptibility fol-lowing phage treatment.

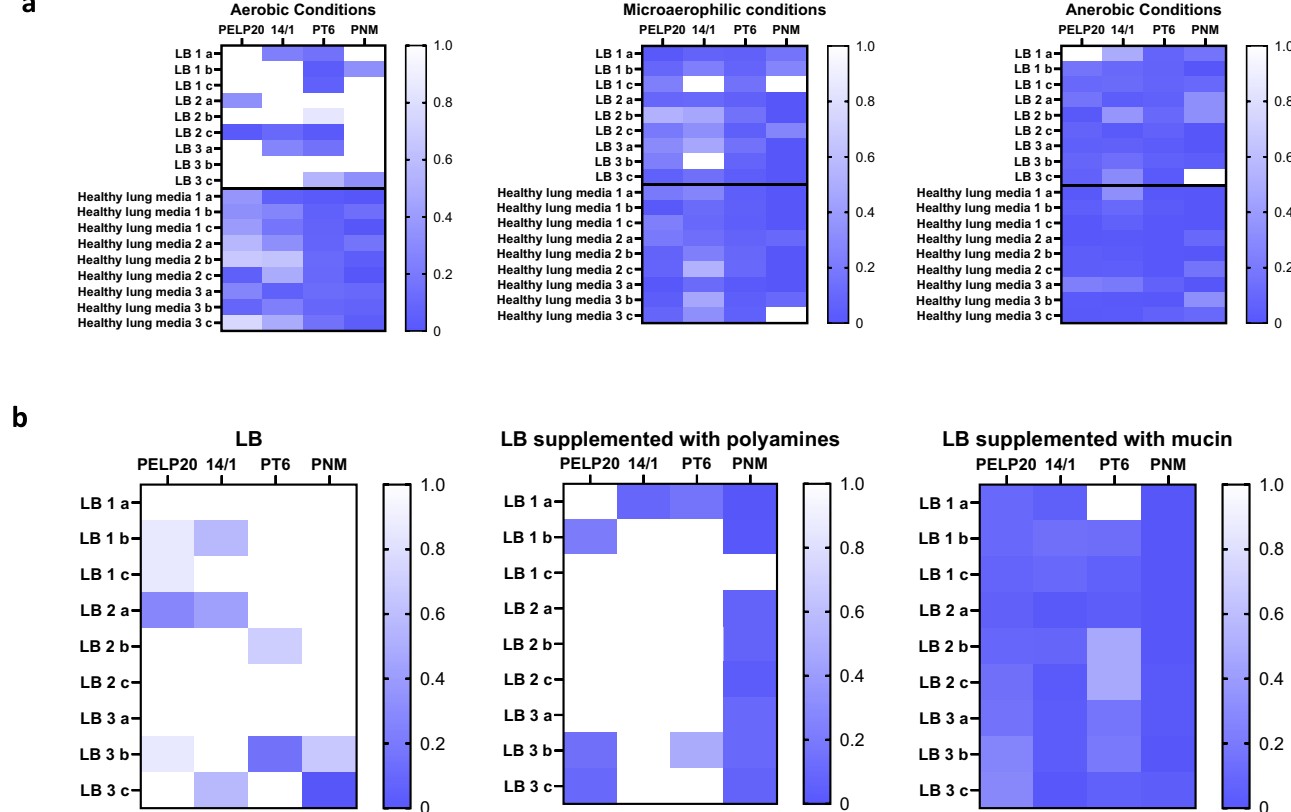

**Fig. 5 | Adaptation to the lung environment results in the development of phage resistance in the absence of phage treatment. a** Heatmaps displaying phage resistance via efficacy of plating (EOP) of isolates recovered from B9 populations incubated for 48 h in either Luria-Bertani (LB) or Healthy Lung Media (HLM) under aerobic, microaerophilic, or anaerobic conditions. Each group and condition include 3 populations (referred to by number), and 3 isolates from each population were tested (referred to by letter). Score of 1 indicates complete susceptibility (white squares), while score of 0 indicates total phage resistance (blue squares). **b** Heatmaps illustrating phage resistance via EOP from B9 populations grown for 48 h in LB, LB supplemented with polyamines (same concentration as HLM), and LB supplemented with mucin (same concentration as HLM). Each group and condition include 3 populations (referred to by number), and 3 isolates from each population were tested (referred to by letter). Score of 1 indicates complete susceptibility (white squares), while score of 0 indicates total phage resistance (blue squares). Source data are provided as a Source Data file.

Timing of phage administration appears to be a key factor in antibiotic sensitisation as only small changes in antibiotic susceptibility were seen in early phage treated isolates (Supplementary Fig. 5)

### Genetic alterations in phage resistant isolates

To further investigate the mechanism of phage resistance and antibiotic re-sensitisation, we conducted whole genome sequencing to compare acquisition of mutations between non phage treated in vivo adapted isolates and the phage treated in vivo adapted isolates. Across all isolates sequenced, there was no evidence of loss of the megaplasmid, and no mutations on the plasmid to indicate altered expression of plasmid-encoded AMR genes.

In isolates recovered from the lungs of non-phage treated mice, we found evidence of a frameshift variant in gene *FC629_24630*, a glycosyltransferase family 2 protein which has homology with PA01 *migA*, an LPS associated alpha-1,5-rhamnosyltransferase[24]. Alterations in LPS biosynthesis could explain the development of resistance in the absence of phage treatment via modification of the phage adsorption receptor for LPS-targeting phages (14/1 and PELP20). There was also evidence of an 810 kb duplication in isolates recovered from the lungs, which could have an impact of phage resistance by altering expression of many genes within this region (Fig. 8, Supplementary data 2). Furthermore, a number of SNPs present at low frequency in the B9 infection stock have become fixed in nearly all the in vivo adapted isolates, suggesting they are advantageous for *P. aeruginosa* survival in vivo (Supplementary Fig. 6, Supplementary

data 2). Overall, there is evidence of *P. aeruginosa* adapting to the in vivo lung environment.

The largest number of gene variants was observed in early treated phage isolates, where phage resistance occurred but limited antibiotic re-sensitisation was observed (Supplementary Figs. 5 and 6). There was evidence of parallel acquisition of mutations targeting *FC629_09380* in isolates recovered from multiple tissues: FC629_09380 is a hypothetical protein which has homology with a PA103 putative O-antigen polymerase, therefore may be involved in LPS biosynthesis. This could explain the development of phage resistance in these isolates via modification of LPS preventing adsorption of LPS-targeting phages (14/1 and PELP20). In early treated isolates recovered from the liver, the same 810 kb duplication event discussed above (in isolate from lung of non-phage treated mice) was observed, in addition to a large (241 kb) deletion event. Such large duplication and deletion events suggest strong evolutionary selection for adaptation to these environments as such mutational events have the capacity to alter expression and regulation of many bacterial functions. Isolates from the kidneys of early phage-treated mice also bore SNPs in both *FC629_24630* and *FC629_09380*, associated with LPS biosynthesis, plus a large (-64 kb) deletion, which could explain the development of phage resistance in these isolates (Figs. 4 and 8, Supplementary Fig 5).

Unexpectedly, in the delayed phage treatment, where broad phage resistance was acquired and re-sensitisation to antibiotics was seen, only two isolates carried evidence of any SNPs: one isolate recovered from the lungs contained a frameshift variant of

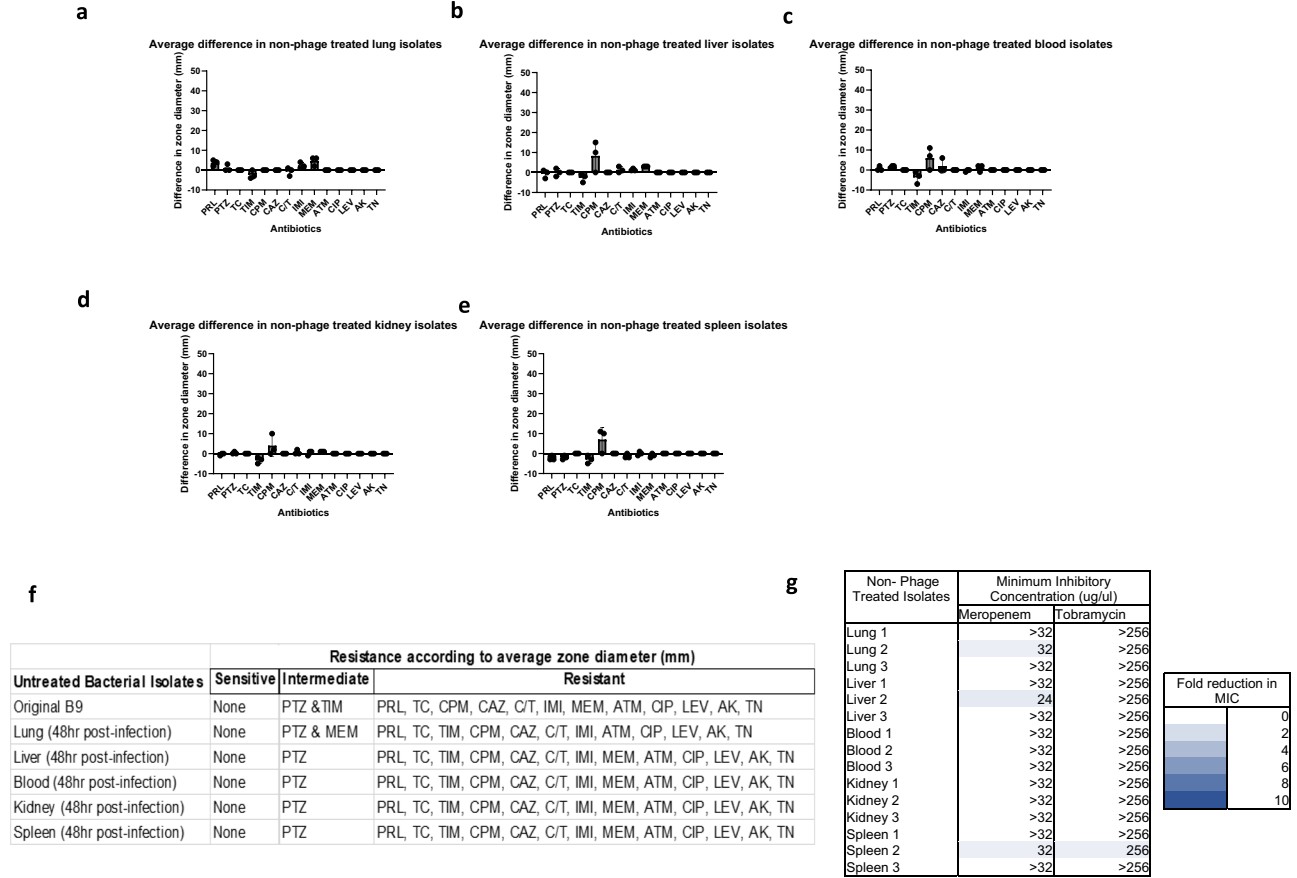

**Fig. 6 | Antibiotic susceptibility of non-phage treated isolates recovered from systemic infection in vivo model.** Difference in inhibition zone diameter in millimetres compared to the input *P. aeruginosa* to a panel of antibiotics for isolates recovered from the **a** lung, **b** liver, **c** blood, **d** kidney and **e** spleen. The mean with SD is indicated and *n* = 3 isolates tested per group **f** shows a summary of changes in resistance classification according to EUCAST breakpoints. **g** Minimum Inhibitory Concentration (MIC) of tobramycin and meropenem determined via E-Test for input and in vivo adapted isolates, with fold reduction in MIC shown using a colour scheme (white: no change, dark blue: 10-fold change). Panel of antibiotics included: Piperacillin (PRL), Piperacillin-tazobactam (PTZ), Ticarcillin (TC), Ticarcillin-clavulanic acid (TIM), Cefepime (CPM), Ceftazidime (CAZ), Ceftolozane-tazobactam (C/T), Imipenem (IMI), Meropenem (MEM), Aztreonam (ATM), Ciprofloxacin (CIP), Levofloxacin (LEV), Amikacin (AK), Tobramycin (TN). Source data are provided as a Source Data file.

*FC629_09380* and an isolate from the kidneys contained a missense variant in *FC629_24630* (Fig. 8). While this may explain LPS-associated phage resistance in these isolates, this does not explain the broad phage resistance (including to the phages PNM and PT6 which do not require LPS for adsorption) and subsequent re-sensitisation to antibiotics in the other delayed phage treated isolates. This suggests that the mechanism of phage resistance is post-transcriptional and differs from the isolates recovered from the early phage treated experiment. It has been shown that clinical strains of *P. aeruginosa* infected with LPS targeting phages, tend to have mutations in regulatory genes rather than LPS biosynthesis genes unlike reference strain PAO1, supporting this hypothesis[25].

### Re-sensitised isolates have increased outer membrane permeability

In delayed phage treated isolates, as we saw evidence of altered antibiotic sensitivity to antibiotics from different classes and no evidence of a loss of antibiotic resistance genes, we decided to investigate whether the outer-membrane permeability of isolates recovered from the lungs was altered compared to non-phage treated isolates. We conducted membrane permeability assays to investigate integrity of the outer membrane (using fluorescent dye propidium iodide, a N-phenyl-1-naphthylamine assay) and to investigate cytoplasmic membrane polarisation (3,3'-dipropylthiadicarbocyanine iodide assay). While we found no significant differences in cytoplasmic membrane polarisation, the delayed phage treated isolates had significantly more permeable outer membranes than either the input B9 isolate or the non-phage treated isolates (Fig. 8b, c).

Overall, this suggests that delayed phage treated isolates were re-sensitised to antibiotics due to post-transcriptional changes which results in the increased permeability of the outer membrane.

### Phage treatment in vivo re-sensitises *P. aeruginosa* to antibiotics

Phage steering is a therapeutic strategy using phages to kill phage sensitive bacteria while 'steering' survivors towards an antibiotic sensitive phenotype. As we observed antibiotic re-sensitisation associated with in vivo evolved phage resistance, we sought to determine whether phage steering could be exploited to improve bacterial clearance within our in vivo model. Two antibiotics were utilised: meropenem and tobramycin. Meropenem is a key carbapenem antibiotic commonly used to treat systemic *P. aeruginosa* infections. Furthermore, isolates treated with phage had been re-sensitised to meropenem, displaying a shift from clinically resistant to susceptible based on EUCAST breakpoints[26] and reduction in MIC. Tobramycin is an aminoglycoside, often used against *P. aeruginosa*. Re-sensitisation was observed with movement from the resistant to the intermediate class, with some isolates displaying >4 fold reduction in the MIC.

Mice were infected with *P. aeruginosa* and treated with phage cocktail at 5 h, then further treated with either meropenem (1 mg/kg) or tobramycin (5 mg/kg) at 48 h post infection. Bacterial loads in the

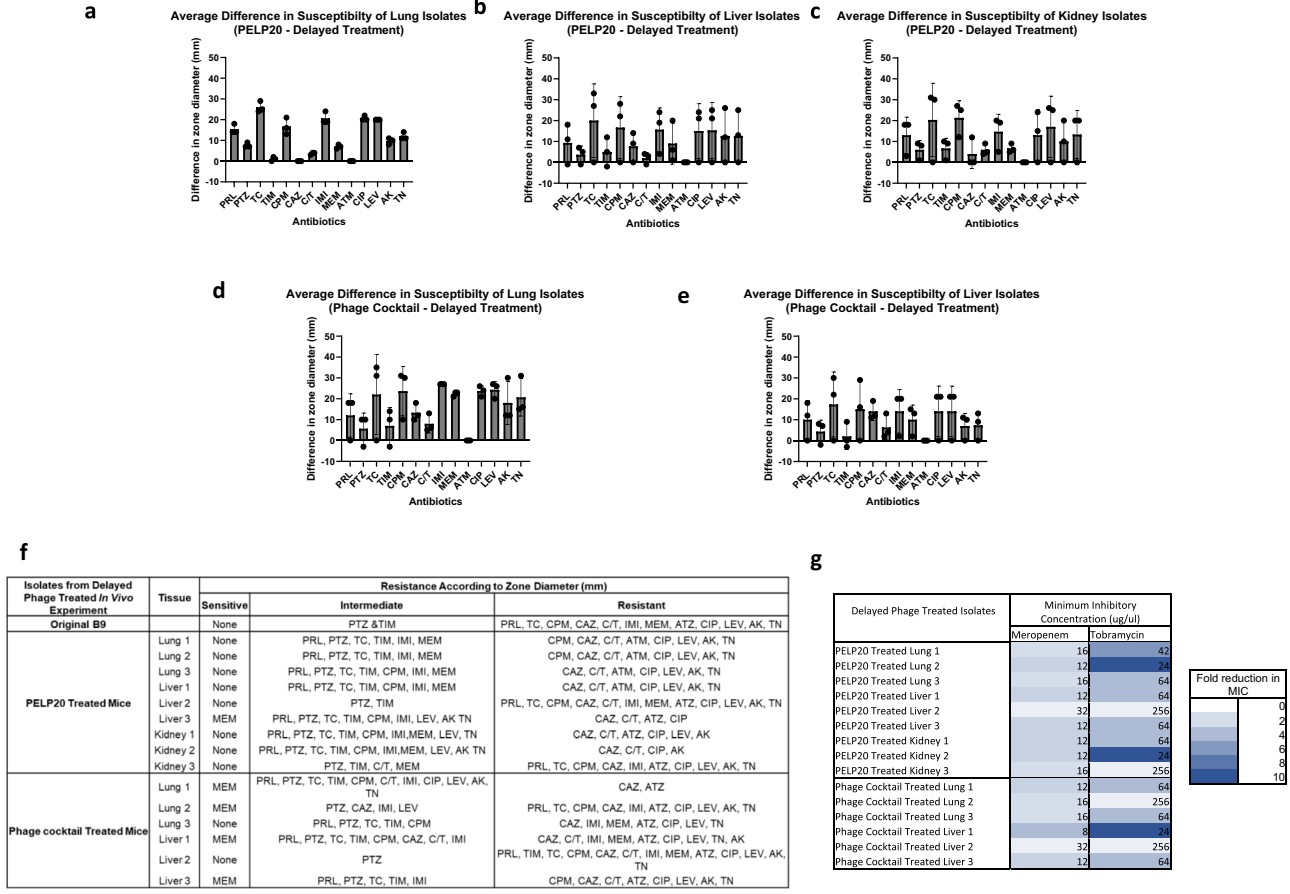

**Fig. 7 | Increased antibiotic susceptibility of delayed phage treated isolates recovered from systemic infection in vivo model.** Difference in inhibition zone diameter in millimetres compared to the input *P. aeruginosa* isolate to a panel of antibiotics for isolates recovered from the **a** lungs, **b** liver, and **c** kidney of delayed PELP20 treated mice and the **d** lungs and **e** liver of delayed phage cocktail treated mice. The mean with SD is indicated and *n* = 3 isolates tested per group **f** shows a summary of changes in resistance classification according to EUCAST breakpoints. **g** Minimum Inhibitory Concentration (MIC) of tobramycin and meropenem determined via E-Test for input and in vivo adapted isolates with fold reduction in MIC shown using a colour scheme (white: no change, dark blue: 10-fold change). Panel of antibiotics includes: Piperacillin (PRL), Piperacillin-tazobactam (PTZ), Ticarcillin (TC), Ticarcillin-clavulanic acid (TIM), Cefepime (CPM), Ceftazidime (CAZ), Ceftolozane-tazaobactam (C/T), Imipenem (IMI), Meropenem (MEM), Aztreonam (ATM), Ciprofloxacin (CIP), Levofloxacin (LEV), Amikacin (AK), Tobramycin (TN). Source data are provided as a Source Data file.

lungs, liver, blood, spleen, and kidney were determined 72 h post-infection. Complete bacterial clearance was achieved following treatment with phage cocktail, then meropenem (Fig. 9). In contrast, meropenem alone did not result in bacterial clearance from all organs but achieved clearance from the blood (Fig. 9c). For phage and tobramycin treatment, bacterial clearance was observed for all organs apart from the liver where bacteria were isolated from 2/10 mice. Compared with tobramycin only treated mice, a statistically significant (*p* = 0.0238) reduction was seen in the lungs of phage cocktail and tobramycin treated mice. In summary, infected mice pre-exposed to phage cocktail re-sensitised pan-resistant *P. aeruginosa* to two clinically relevant antibiotics, enabling improved clearance of bacterial load. This highlights the potential therapeutic impact of sequential phage-antibiotic therapy for treating highly antibiotic resistant infections.

In order to determine whether the increased efficacy of phage cocktail and antibiotic therapy was due to phage exposure leading to antibiotic re-sensitisation of bacteria, the treatments were administered in the reverse order (antibiotic first, followed by phage). Mice were infected with *P. aeruginosa* and treated with either meropenem (1 mg/kg) or tobramycin (5 mg/kg) at 5 h, then treated with phage cocktail at 48 h. Bacterial loads in the lungs, liver, blood, spleen, and kidney were determined 72 h post-infection. Administration of antibiotics before the phage cocktail had limited therapeutic success

(Fig. 10). Complete bacterial clearance was only achieved in the blood (and spleen of mice treated with tobramycin then phage cocktail) (Fig. 10c, d). Additionally, no significant differences were seen between any of the treatment groups. This emphasises the importance of administering phage cocktail *before antibiotics* to exploit phage steering in a clinical setting.

## Discussion

Using a clinically relevant in vivo model of systemic infection, we successfully cleared a pan-resistant *P. aeruginosa* infection by exploiting phage steering. Phage cocktail alone had strong therapeutic potential in vivo, regardless of timing or administration route, by significantly reducing or clearing pan resistant *P. aeruginosa* from various tissues. However, phage resistance developed in the lungs even in the absence of phage exposure, with evidence that bacterial adaption to the lung environment leads to phage resistance as a secondary effect, which to our knowledge has not been observed in vivo previously. Moreover, in some tissue sites, low numbers of bacteria recovered from phage treated mice developed resistance to the input phages via reduced adsorption but lacked modification of phage receptors. Interestingly, these phage resistant isolates had remarkably increased antibiotic susceptibility across multiple classes of antibiotic due to altered outer membrane permeability, compared with the

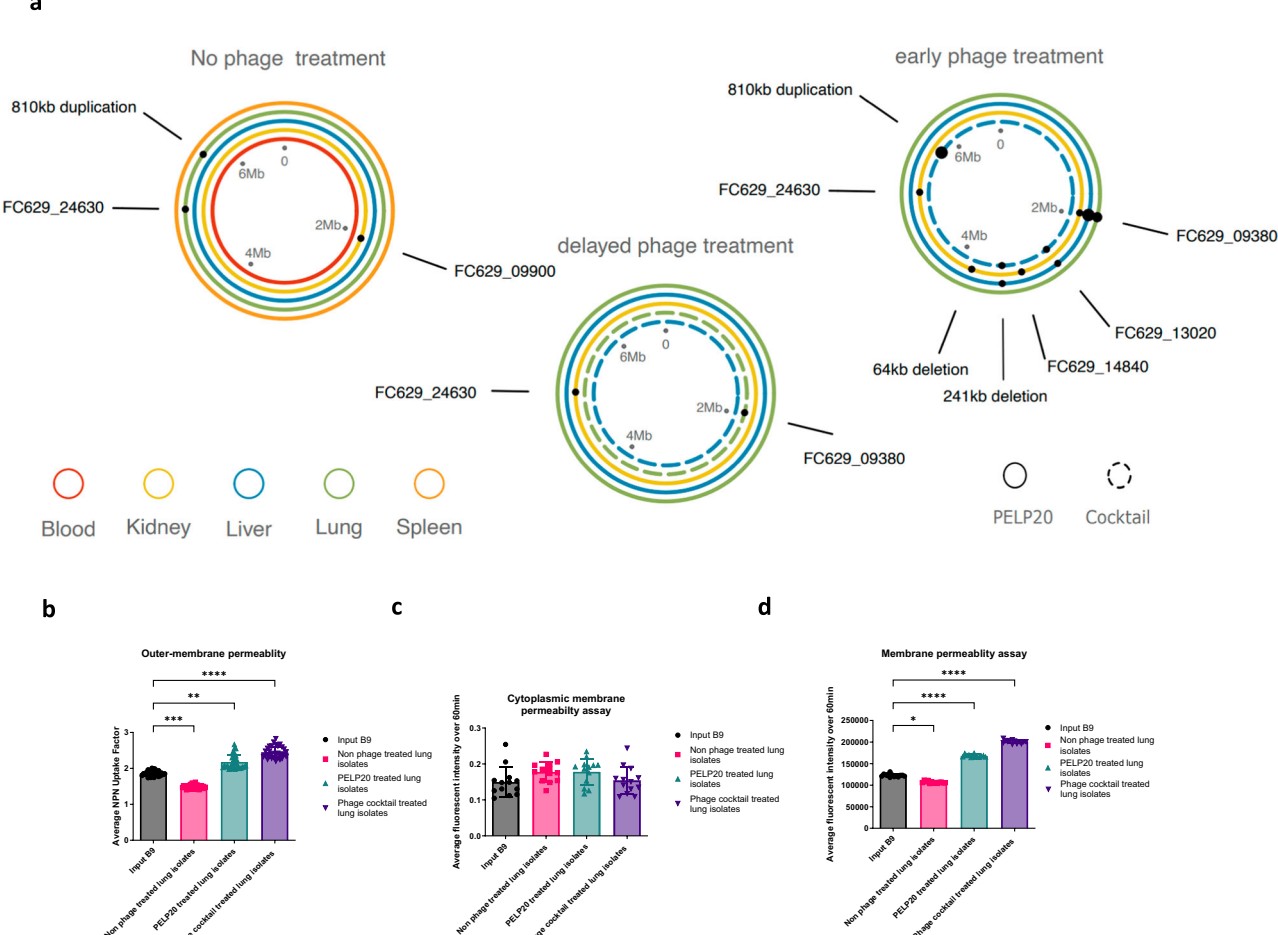

**Fig. 8 | Genetic variation and outer membrane permeability of non-phage and phage treated in vivo adapted isolates. a** Validated gene variants in non-phage treated, early phage treated, and delayed phage treated isolates that were not present in the input B9 ancestor. Lines represent bacterial genomes from different tissues. Solid line: PELP20 treated; dashed line: phage cocktail treated. Dots indicate variants, with size showing the number of isolates (1–3) containing each variant. **b** Membrane permeability measured by propidium iodide fluorescence for input B9 isolate (black circles), with significant differences compared to in vivo adapted isolates from lungs (pink squares, $p = 0.0312$), delayed PELP20 isolates from lungs (green triangles, $p < 0.0001$), and delayed phage cocktail isolates from lungs (purple triangles, $p < 0.0001$). Mean with SD is indicated, symbols represent fluorescence readings over 1 h, $n = 3$. **c** Outer membrane permeability measured by 1-N-phenylnaphthylamine (NPN) uptake factor of the input B9 isolate (black circles), with significant differences compared to in vivo adapted isolates from lungs (pink squares, $p = 0.0002$), delayed PELP20 isolates from lungs (green triangles, $p = 0.0017$), and delayed phage cocktail isolates from lungs (purple triangles, $p < 0.0001$). Mean with SD is indicated, symbols represent fluorescence readings over 15 mins, $n = 3$. **d** Cytoplasmic membrane polarisation measured by 3,3′-dipropylthiadicarbocyanine iodide fluorescence for the input B9 isolate (black circles), with no significant differences compared to in vivo adapted isolates from the lungs (pink squares), delayed PELP20 isolates from the lungs (green triangles), and delayed phage cocktail treated isolates (purple triangles) from the lungs. Mean with SD is indicated, symbols represent fluorescence over 1 h, $n = 3$. Statistics were performed using a two-way ANOVA with Bonferroni correction ****$p < 0.0001$ *** $p < 0.001$ **$p < 0.01$ *$p < 0.05$. Source data are provided as a Source Data file and Supplementary data file 2.

input B9 (T2436) isolate, which could then be exploited using phage steering.

Phage steering utilises phages to kill the majority of phage sensitive bacteria while driving survivors towards an antibiotic sensitive phenotype[18–20]. Here, re-sensitisation to antibiotics was achieved rapidly in vivo, leading to clinical susceptibility in a previously carbapenem-resistant strain. Carbapenem resistance is key concern for the WHO, as there has been a worldwide increase in carbapenem resistant hospital acquired infections[1]. Previous in vitro work highlighted the potential for phage steering to treat MDR infections in *Acinetobacter baumannii* and *P. aeruginosa*[18–20]. However, it was unknown if phage steering can elicit antibiotic re-sensitisation in vivo, a key point shown in this study.

The concept of phage steering relies on the development of phage resistance and subsequent sensitivity to antibiotics. Phage resistance evolved within our in vivo model and, intriguingly, phage resistance developed in the lungs even in the absence of phage treatment. While variable levels of phage resistance were seen in other tissues in the absence of phage treatment, the lungs were the only niche with near total resistance to the input phages. To our knowledge, no other study has observed niche specific resistance in vivo. In contrast, phage treated isolates from the lungs were susceptible to input and in vivo adapted phages, suggesting that bacteria and phage adapt to the in vivo environment. *P. aeruginosa* is known to alter its outer surface proteins when adapting to an in vivo environment[27] and bacterial adaptation to the in vivo environment limits phage efficacy[28]. We also show that *P. aeruginosa* exposed to factors found within the lung environment in vitro (limited oxygen availability, mucin and polyamines), promotes phage resistance in the absence of phage exposure. It has been shown that *P. aeruginosa* grown in oxidative stress have altered lipopolysaccharide formation in their outer-membrane[13], which supports our genetic analysis where non-phage treated isolates

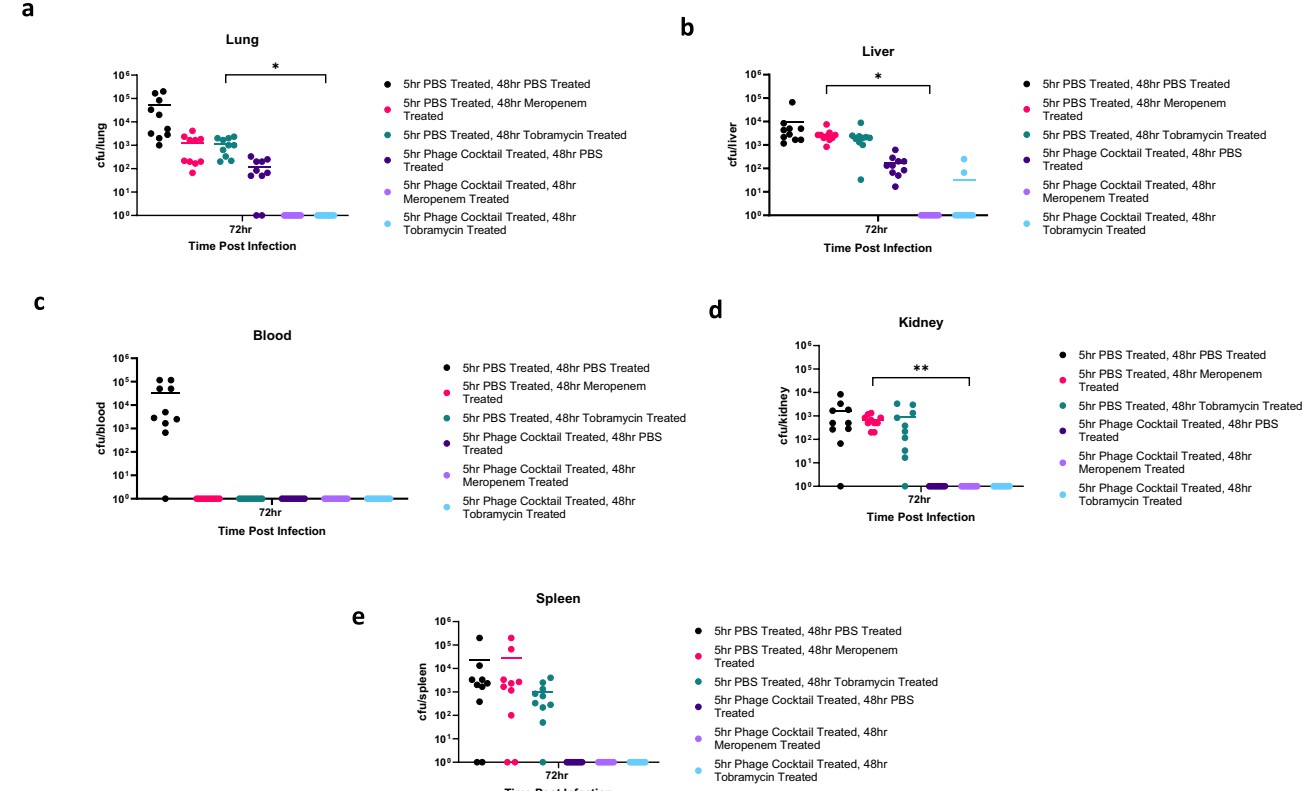

**Fig. 9 | Pre-exposure to phage cocktail results in re-sensitisation to antibiotics in vivo.** Bacterial colony forming units (CFU) in *P. aeruginosa* infected mice after treatment with either PBS or phage cocktail at 5 h post bacterial infection, and treatment with either phosphate buffered saline (PBS), meropenem, or tobramycin at 48 h post bacterial infection. **a**–**e** CFU in the lung, liver, blood, kidney, and spleen 72 h post bacterial infection. Treatment groups include PBS (black), phage cocktail (dark purple), tobramycin (green), meropenem (pink), phage cocktail and meropenem (light purple) and phage cocktail and tobramycin (light blue). Each symbol represents an individual mouse and line indicates the mean. Results are combination of two independent experiments, $n = 10$ mice per group per time-point. Significant difference in CFU between PBS & tobramycin and phage cocktail & tobramycin treated mice was observed in the lung ($p = 0.0238$). Significant differences in CFU between PBS & meropenem and phage cocktail & meropenem treated mice were seen in the liver ($p = 0.0133$) and kidney ($p = 0.0046$). The y-axis has been corrected by adding 1 (to zero all samples). Statistics were performed using a one-way ANOVA Bonferroni correction **$p < 0.01$ *$p < 0.05$. Source data are provided as a Source Data file.

from the lung contained a frameshift variant in gene *FC629_24630*, a homologue of PA01 *migA*, a LPS biosynthesis protein. This suggests that some niches will be more difficult to treat with phages, having important implications for clinical application. Furthermore, *P. aeruginosa* displays both genetic and phenotypic heterogeneity, thereby potentially making chronic lung infections a greater phage treatment challenge.

Bacteria recovered from phage treated mice were resistant to subsequent phage infection due to reduced adsorption. Bacteria can develop phage resistance by modifying the targeted surface receptor preventing phage infection, however mutations in surface receptors can lead to fitness costs[29]. Phage resistant bacteria can become re-sensitised to other phages, antibiotics, and the complement system[18–20]. Similarly, we observed previously pan-resistant isolates from phage treated mice were re-sensitised to a number of different antibiotics including carbapenems due to increased outer membrane permeability. However, while we saw genetic evidence to explain the development of phage resistance in some isolates, we saw no genetic basis for the altered antibiotic sensitivity suggesting that this mechanism is epigenetic/post transcriptional. Further work on transcriptomes of these isolates would need to be conducted to gain further understanding on how phage treatment can lead to altered antibiotic sensitivity, including the potential impact on tolerance and persistence. Therefore, understanding the impact of predator-prey dynamics on the efficacy of phage therapy in vivo is important.

Although complete bacterial clearance was achieved via phage steering, phage cocktail alone had strong therapeutic potential in vivo. However, efficacy of phage treatment and the development of resistance differed between biological niches. This may also be due to phage pharmacodynamics as phage penetration varied in different niches. Within mammalian systems, clearance of phages is primarily via the reticuloendothelial system (RES)[30,31]. The liver and spleen are most associated with the accumulation of bacteriophages[32,33], with the liver particularly associated with phage accumulation and clearance, as 90% of the RES is located in the liver[34]. In contrast, phages tend to be present at lower concentrations in the kidneys[35,36]. This may explain how phages cleared the spleen in our in vivo model and why phage resistance varied between tissues.

While the phage cocktail achieved high levels of efficacy, a high concentration of each phage was needed and as such the phage used in this study were not purified to remove endotoxins, a process which often reduces the phage titre. Before being able to be used clinically, this cocktail would need to undergo various purification steps to be safe to administer. Additionally, while the phage cocktail was found to be more efficacious than single phage treatment, the overall concentration of phages within the cocktail was higher ($1 \times 10^{12}$ pfu/ml) vs PELP20 alone ($5 \times 10^9$ pfu/ml). While this provides evidence that the other phages improve efficacy, it would be interesting to see if administering PELP20 at $1 \times 10^{12}$ would achieve the same efficacy as the phage cocktail, to further explore phage-phage interactions.

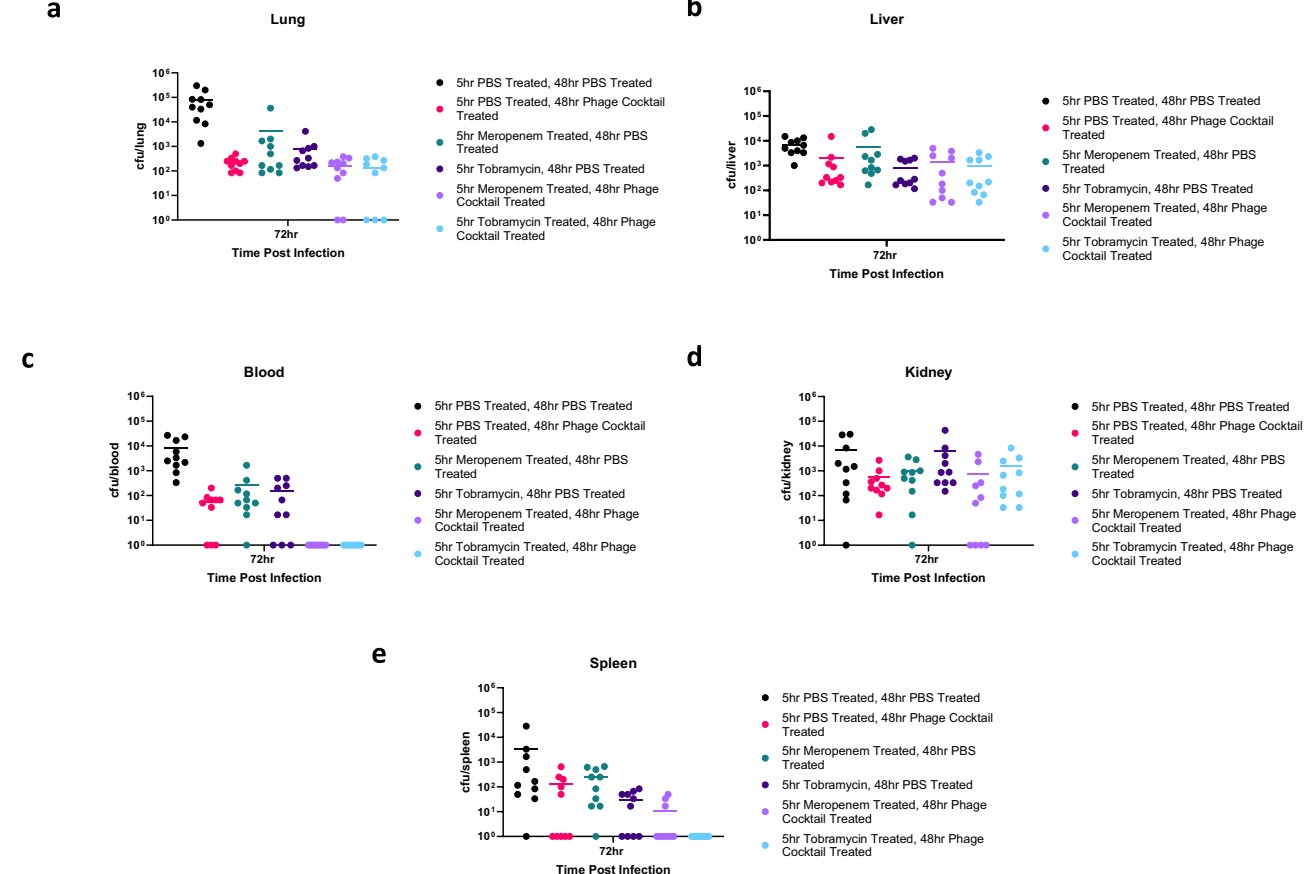

**Fig. 10 | Administration of antibiotics before phage cocktail has limited therapeutic effects.** Bacterial colony forming units (CFU) in *P. aeruginosa* infected mice after treatment with either phosphate buffered saline (PBS), meropenem or tobramycin at 5 h post bacterial infection, and treatment with either PBS or phage cocktail at 48 h post bacterial infection. **a**–**e** CFU in the lung, liver, blood, kidney, and spleen 72 h post bacterial infection. Treatment groups include PBS (black), phage cocktail (pink), tobramycin (dark purple), meropenem (green), phage cocktail and meropenem (light purple) and phage cocktail and tobramycin (light blue). Each symbol represents an individual mouse and line indicates the mean. Results are combination of two independent experiments, *n* = 10 mice per group per timepoint. The y-axis has been corrected by adding 1 (to zero all samples). Statistics were performed using a one-way ANOVA, Bonferroni correction comparing mock treated groups and both treatment groups, no significant differences were observed. Source data are provided as a Source Data file.

Phage treatment reduced *P. aeruginosa* infection regardless of administration route. However, intravenous administration of phage cocktail was significantly more effective than delayed intranasal administration against systemic *P. aeruginosa* infection (Fig. 3 & Supplementary Fig. 3). In a recent study, the efficacy of intranasal phage therapy in a *P. aeruginosa* pneumonia murine model reduced when administration was delayed from 2 h to 8 h post infection[37]. Additionally, intravenous administration of phage therapy was more effective than intranasal administration in a *Burkholderia cenocepacia* pulmonary infection murine model[38]. We hypothesise that reduced systemic efficacy of delayed intranasal administration is due to lower numbers of phage translocating into blood post-inhalation. We show efficacy of phages against *P. aeruginosa* in blood post-delivery to lungs, albeit with reduced impact across other tissues (Supplementary Fig 3). In contrast, intravenous phage administration was clearly more effective at reducing CFU in other organs (Fig. 3). After intravenous administration, phages rapidly disseminate to other tissues, reaching the spleen and liver within minutes[34]. Clearly, timing and administration route has an important impact on the efficacy of phage therapy. Future studies tracking the dynamics of individual phages within a cocktail would provide even greater insight into the therapeutic potential of particular phage for different infection types.

The host range for our phage cocktail covered a broad range of *P. aeruginosa* isolates. These included isolates from patients with keratitis, hospital acquired infections, and urinary tract infections. Therefore, our phage cocktail could potentially treat multiple types of *P. aeruginosa* infections. However, we conducted our initial screen using direct spot test, and it has been shown previously this method can overestimate the efficacy of a phage due to phenomena "lysis from without" where bacterial death is caused by high phage adsorption or lysins within the phage preparation rather than true infectivity[39]. Additionally, while we tested a broad range of bacterial isolates, the vast majority were keratitis isolates, to determine the host range more isolates from hospital acquired infections and UTIs could have been included.

In summary, the potential to use phage therapy to reduce the bacterial burden of pan-resistant *P. aeruginosa* and simultaneously re-sensitise bacteria to antibiotics is game changing. Therefore, sequential administration of bacteriophage and antibiotic therapy could be a viable solution in combating pan-resistant *P. aeruginosa* infections.

## Methods
### Ethical approval
All experimental protocols involving animals were approved and performed in accordance with the regulations of the Home Office Scientific Procedures Act (1986), project licence P86De83DA and the University of Liverpool Ethical and Animal Welfare Committee.

### Direct spot test assay

Bacteriophages PELP20, PT6, PMN and 14/1 (described in Supplementary Table 1) were screened for activity against 551 clinical bacterial isolates using direct spot assays. These included 512 keratitis isolates, 15 UTI isolates and 3 burn wound isolates from the UK, 7 UTI isolates from Kuwait and 12 multidrug resistant isolates from Thailand (Supplementary Table 1). These isolates were grown in LB broth at 37 °C, 180 RPM. Clinical isolates were sourced from previously published studies[21].

The screen for phage activity was determined via direct spot test, 10ul of each phage were spotted on LB plates with an overlay of each clinical isolate, and left overnight at 37 °C. The following day, the phage activity against each isolate was recorded as sensitive (+) donating clear zone of inhibition, (±) indicating turbid or fainter zone of inhibition and sensitive (−) to indicate resistance (Supplementary Data 1).

### Preparing bacteriophage cocktail

We constructed a phage cocktail containing four bacteriophages: PELP20, PT6, PMN and 14/1, described in Supplementary Table 1. The bacteriophage PELP20 has been shown to be effective in vivo against Liverpool Epidemic *P. aeruginosa* strains[12]. Two of the other phages present in the cocktail (PNM and 14/1) have been previously included in a well-defined cocktail available for use in clinical human trials, and all three were shown to be effective against the input B9 (T2436) strain[21].

The appropriate MOI was determined against B9 for each individual phage before assembling the cocktail via checkerboard assay. *P aeruginosa* strain B9 (T2436) was grown in LB broth at 37 °C, 180 RPM. Overnight cultures of bacteria were diluted in LB Media to an optical density (OD) measured at 600-nm wavelength (OD600) of 0.05 (±0.01). Fresh LB Media was inoculated 1:100 with diluted overnight cultures and 100 ul of culture were grown in 96-well micro-titre plates for 24 h at 37 °C with 50 ul of PELP20 and 50 ul of either 14/1, PT6 or PNM at different MOI in a checkboard fashion. The O.D 600 was determined using Flurostar Omega plate reader. Evidence of an additive effect was seen with PELP20 with each phage (Supplementary Fig. 7).

Phages were propagated on strain PA01 lawns via double-layer agar method. Phages were collected by adding 10 ml of PBS to the double agar plates and left to incubate for 5 h. Then the lysate was filtered using a 0.22 uM filter to remove any live bacteria. Phage concentration was determined by plating serial dilutions of each phage onto bacterial PA01 lawns via double-layer agar method, where phage titres were then enumerated. Each phage PELP20, PNM, PT6, and 14/1 were diluted to the appropriate PFU/ml with PBS. The bacteriophage cocktail was then made up with equal volume of each phage in the following concentrations (chosen based on in vitro MOI and starting inoculation of $1 \times 10^6$ cfu/ml): PELP20 MOI $5 \times 10^4$, PNM MOI $5 \times 10^5$, PT6 MOI $5 \times 10^4$, 14/1 MOI $5 \times 10^5$. This resulted in phage concentrations of PELP20 ($5 \times 10^9$ pfu), PNM ($4.95 \times 10^{12}$ pfu), PT6 ($5 \times 10^9$ pfu), 14/1 ($4.95 \times 10^{12}$ pfu). Therefore, overall starting concentration of $1 \times 10^{12}$ pfu for phage cocktail administered to mice.

### Mice

For all mouse experiments, 6–8-week-old female *BALB/c* mice (Charles River) were used and allowed to acclimatise for 7 days prior to use under conditions described previously[12]. The animals were housed in the animal facilities under the following conditions; temperature was 21–23 °C and humidity set at 55–65%, 12 h light-dark cycle. Mice were placed in individually ventilated cages (IVC) from Technoplast (GM500). Automatic watering provided reverse osmosis water sterilised by UV radiation and enrichment included nesting material, balcony, dome home and handling tunnel. Mice were randomly assigned to a cage (experimental group) on arrival at the unit by staff with no role in study design.

### Systemic *P. aeruginosa* mouse infection model

Mice were anesthetised with a mixture of $O_2$ and isoflurane and challenged intranasally with *P. aeruginosa* at $1 \times 10^6$ cfu/ml in 50 ul of PBS. Infection stocks were prepared by centrifuging an 1 ml of overnight culture of *P. aeruginosa* at 140000 RCF and resuspending in 20 ml of Tryptan Soy Broth supplemented with 10% foetal bovine serum and growing for 5 h at 37 °C shaking 180 RPM. 1 ml aliquots of the culture were prepared and stored at −80 °C. Mice were culled as described previously[12], and the lungs, liver, blood, kidneys, and spleen were dissected, homogenised and bacterial load was established via colony forming units (CFU)[12] on *Pseudomonas* selective agar.

For the preliminary survival experiment, we tested four different strains of multidrug resistant *P. aeruginosa* isolated from clinical sputum samples from patients in Thailand; these included strains B3 (T2101), B8 (T2584), B9 (T2436), and C7 (T3582) (5 mice per group per timepoint) (Supplementary Fig. 1). Mice challenged as described previously and were periodically monitored for clinical signs of disease. Mice were sacrificed at 48 h post-infection and the bacterial load in the lungs, liver, blood, kidneys, and spleen was established. In all subsequent experiments, mice were challenged with strain B9 at $1 \times 10^6$ CFU/ml.

To further characterise the model, mice were challenged with B9 (T2436) and sacrificed at established time points (6 h, 12 h, 24 h and 48 h) (5 mice per group per timepoint). The lungs, liver, blood, kidneys, and spleen were dissected and colony forming units were assessed. An additional set of mice were also infected, and tail bled at 2 h and 4 h (Supplementary Fig. 2).

### Early and delayed administration of phage cocktail in mouse model

For the early administration model, 10 mice per group per timepoint were challenged with strain B9 (T2436) in two independent experiments, as previously described. Then immediately after infection, mice were treated intranasally in a 50 ul dose of either PBS (mock treated), PELP20 ($5 \times 10^9$ pfu) or phage cocktail PELP20 ($5 \times 10^9$ pfu), PNM ($4.95 \times 10^{12}$ pfu), PT6 ($5 \times 10^9$ pfu), 14/1 ($4.95 \times 10^{12}$ pfu). Mice were periodically monitored for clinical signs of disease and culled at established timepoints (6 h, 12 h, 24 h and 48 h post infection). The lungs, liver, blood, kidneys, and spleen were dissected and colony forming units were assessed.

For the delayed administration model, mice were infected as described previously in the early infection model (10 mice per group per timepoint in two independent experiments). Then 5 h post-infection mice were treated intravenously in a 50 ul dose with either PBS (mock treated), PELP20 or phage cocktail. Mice were culled and tissues were dissected and processed same as the early administration experiment.

At 48 h, bacterial isolates (three per treatment group per tissue site) were harvested from any organs where bacteria could be cultured. For both early and delayed experiments, bacterial isolates were harvested in triplicate from mock-treated, PELP20 treated and phage cocktail treated mice and stored as 25% glycerol stocks at −80 °C.

### Pharmacokinetic analysis of phage cocktail in mouse model

The concentration of phage cocktail in the tissues was determined for the early and delayed experiments. Mice were infected and treated with phage cocktail (5 mice per group) as described previously. Additionally, mice mock infected with 50 ul of sterile PBS were treated with phage cocktail as described previously. Mice sacrificed 48 h post-infection and concentration of phage was established via spotting serial dilutions onto bacterial lawns of input B9 strain via the double agar method[39].

## Antibiotic sensitivity assays

A panel of clinically relevant antibiotics were selected to conduct disk diffusion assays. Each isolate harvested from mice sacrificed 48 h post-bacterial infection was grown on Mueller–Hinton broth overnight at 37 °C, 180 RPM and then disk diffusion assay was performed[23]. Breakpoints were established according to the latest EUCAST guidelines[23].The minimum inhibitory concentration (MIC) of antibiotics meropenem (32 ug/ml) and tobramycin (257 ug/ml) was determined via E-test strips provided by bioMérieux[40]. Overnight cultures of each isolate we adjusted to a McFarland standard of 0.5 and each inoculum was applied to a Mueller–Hinton Agar plate with a sterile cotton swab and allowed to dry for 10 min. E-TEST strips were then placed on inoculated plates and the plates were incubated for 18–20 h at 37 °C prior to reading the MIC.

## Phage sensitivity assays

Phage resistance was determined by efficiency of plating[41].For each bacterial isolate, bacterial lawns was made using the double-layer agar method. Serial dilutions of each phage were spotted onto the bacterial lawn, where phage titres were enumerated and the EOP of each recovered isolate compared to the original B9 isolate was established.

## Phage cocktail and antibiotic therapy in mouse model

Mice were infected as described previously, 10 mice per group in two independent experiments. Then mice were treated intravenously 5 h post infection with either phage cocktail as described previously, or with sub-inhibitory dose of meropenem (1 mg/kg) or tobramycin (5 mg/kg). Then at 48 h post-infection phage-treated mice were given either tobramycin or meropenem and antibiotic treated mice were given phage cocktail intravenously. Bacterial loads in the lungs, liver, blood, spleen, and kidney were determined 72 h post-infection.

## Bacterial genome sequencing and analysis

Bacterial genomes were sequenced using Illumina platform (MiSeq) to produce paired short reads (-250 bp). The project accession is PRJEB67471. Paired reads were aligned to a *P. aeruginosa* B9 reference genome (GenBank accessions NZ_CP039988 and NZ_CP039989) using Burrows-Wheeler Aligner[42]. Variants (SNPs and small indels) were called using the GATK Haplotype Caller[43] and annotated to determine gene targets and putative effect using SNPeff[44]. Initial filtering of variants to remove low quality calls was performed in R (version 4.0.4[45]), to select variants supported by a read depth of >20 reads per base pair and an allele frequency of >50%. Large duplication and deletion events were detected by analysis of read coverage performed in R (version 4.0.4[45]). Final validation of all variants was performed visually using an alignment viewer (igv[46]), including comparison to re-sequenced ancestral reference to remove likely sequencing errors and pre-existing differences between the ancestral strain and the published reference genome. A number of borderline variants were identified, which were identified at low frequency in the ancestral population, but become fixed (i.e., reached allele frequency of 100%) in a subset of populations (Supplementary Fig. 6, Supplementary data file 2)

## Evolutionary experiment in Healthy Lung Media and LB Broth

Healthy Lung Media (HLM) was prepared as described here[23]. Three overnight culture of *P aeruginosa* strain B9 (T2436) was grown in LB broth at 37 °C. Overnight cultures of bacteria were diluted in LB Media to an optical density (OD) measured at 600-nm wavelength (OD600) of 0.05 (±0.01). Fresh LB Media or HLM was inoculated 1:100 with diluted overnight cultures for a final volume of 3 ml and incubated for 48 h at 37 °C in either aerobic conditions, in a Thermo Scientific™ Oxoid 2.5 L candle jar (microaerophilic conditions), or under anaerobic conditions using Thermo Scientific™ Oxoid AnaeroGen 2.5 L Sachet are anaerobic gas generating sachets for use with Thermo Scientific™ Oxoid 2.5 L jar (anaerobic conditions). At 48 h, populations

(3 biological replicates for each condition) were streaked out on LB agar plates and incubated overnight at 37 °C. Then 3 clones per biological replicate were picked to perform EOP assays for phages PELP20, 14/1, PNM, and PT6 as described previously.

This experiment was repeated for LB broth, LB broth supplemented with polyamines (spermidine-200ng/ml, spermine-32.5 ug/l, and putrescine-616 ug/l) and LB broth supplemented with mucin (1.2 mg/ml), the same concentrations used in HLM under aerobic conditions[23]. Isolates recovered from the LB broth, HLM, LB supplemented with polyamines and LB broth supplemented with mucin were also tested for antibiotic sensitivity via E-TEST for meropenem and tobramycin, with little to no re-sensitisation seen (Supplementary Fig. 8).

## Membrane permeability assays

For the input B9 isolate, non-phage treated isolates recovered from the lungs, and delayed phage treated isolates recovered from the lungs, three assays were performed to determine the outer membrane integrity, cytoplasmic depolarisation and membrane integrity of strain B9, non-phage treated isolates recovered from the lungs, and delayed phage treated isolates recovered from the lungs, three different assays were performed[47].

Integrity of the outer bacterial membrane was measured using the fluorescent probe 1-N-phenylnaphthylamine (NPN; Merck). Late-exponential-phase cultures of each isolate tested were washed twice and adjusted to an OD600 of 0.5 in 5 mM HEPES buffer. In the wells of a black microtiter plate, adjusted cultures were combined with NPN (final concentration, 10 µM) to a final volume of 200 µL. Fluorescence was measured in a Omega BMG plate reader at excitation of 355 nm, emission at 460 nm every 30 s for 15 min. NPN uptake factor was calculated as follows: [(fluorescence of sample with NPN) − (fluorescence of sample without NPN)]/[(fluorescence of buffer with NPN) − (fluorescence of buffer without NPN)].

Depolarisation of the cytoplasmic membranes was measured using the fluorescent probe DiSC3(5) (Thermo Scientific). Late-exponential-phase cultures of each isolate were washed twice and adjusted to an OD600 of 0.05 in 5 mM HEPES–20 mM glucose. DiSC3(5) was added to the bacterial cultures to a concentration of 1 µM and aliquoted to the wells of a black microtiter plate for a final volume of 200ul. The fluorescent signal of the dye was allowed to quench for 30 min in the dark. Fluorescence was measured in a Omega BMG plate reader at excitation of 544 nm, emission at 620 nm every 30 s for 1 h.

Permeabilization of the bacterial membranes of each isolate was measured using the fluorescent dye propidium iodide (PI). Late-exponential-phase cultures of each isolate were washed twice and adjusted to an OD600 of 0.5 in PBS. PI was added to the bacterial cultures to a concentration of 1 µg/mL for a final volume of 200 ul in a black microtiter plate. Fluorescence was measured in a Omega BMG plate reader at excitation at 544 nm, emission at 610 nm every 30 s for 1 h.

## Phage adsorption assay

The phage adsorption rate was determined for the phage cocktail against input B9 isolate, non-phage treated isolates and delayed phage treated isolates in vitro. First each isolate was incubated with phage cocktail at MOI of 1. Then at 5 min, 10 min, 15 min, 30 min, 45 min and 60 min, 500 ul was removed and passed through a 0.22 µM filter to remove bacterial cells, then plaque assays were performed to establish PFU[18].

## Statistics

All statistical analysis were carried out using GraphPad Prism 7 software (GraphPad Inc, La Jolla, Calif), unless otherwise stated. A two-way ANOVA with Bonferroni correction post hoc test was performed when comparing more than 3 experimental groups.

## Reporting summary

Further information on research design is available in the Nature Portfolio Reporting Summary linked to this article.

## Data availability

The genetic sequences acquired during this study have been deposited into the National Center for Biotechnology Information database as a BioProject under accession number PRJEB67471. *P. aeruginosa* B9 reference genome can be accessed via GenBank accessions NZ_CP039988 and NZ_CP039989. Accession number for phage 14/1 is NC_011703 and sequences for PELP20 and PNM under restricted access for privacy reasons, access can be obtained upon request. The data generated in this study are provided in the Supplementary Information and Source Data files. Source data are provided with this paper.

## Code availability

The study did not involve the creation of novel code for bioinformatics analyses. All the used software is freely accessible and is referenced in the Methods section.

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

## Acknowledgements

This work was funded by Medical Research Council Discovery Medicine North Doctoral Training Program, Cystic Fibrosis Trust and CF Foundation funded Strategic research Centre (PIPE-CF), SRC022. J.P.J.H. is supported by an Medical Research Council, UK (MRC) Career Development Award, (MR/W02666X/1).

## Author contributions

E.A, A.K and J.F conceived the study. E.A and R.S. performed in vivo phage treatment experiments. E.A. performed in vivo phage antibiotic treatment experiments. R.W. and JPH performed bioinformatics analyses of bacterial genomes. E.A. performed the in vitro characterisation of phage-resistance, antibiotic sensitivity tests, experimental evolution, and outer membrane permeability assays. J.W. and A.H. conducted the initial direct spot screen assays on the clinical isolates. J.F and A.K. provided resources for experimental work. J.F. and A.K. supervised and funded the project. E.A. and J.F. wrote the original draft. All authors were involved in reviewing and editing the final manuscript.

## Competing interests

The authors declare no competing interests.
