## [Peer Review File · Nature Communications]

REVIEWER COMMENTS

Reviewer #1 (Remarks to the Author):

This manuscript studies the potential of phages for the treatment of pan-resistant *P. aeruginosa* infections using an in vivo model. The authors further study the re-sensitization to antibiotics caused by phage exposure. Although the manuscript is clear, well written and interesting results were achieved, it is long known that phages can revert antibiotic resistance due to mutations that occur in bacteria after phage exposure and so, the concept is not new. There are also some other concerns described below:

- Lines 110-119, line 413 and Table S1: The spot assay is not the appropriate method to assess the infectivity and host range of phages and can lead to false positives. The methodology here is not ok and so, the conclusions based on this assay cannot be drawn. The spot test always has to be complemented with Efficiency of plating assays. If a positive result is observed in the spot test (positive result), then serial dilutions have to be plated to see if individual phage plaques are observed in the bacterial lawn. A positive result in the spot test does not mean that the phage is able to infect and replicate in that particular strain. Phage plaques have to be observed to be able to draw that conclusion. In this case, the Authors did not perform the serial dilutions and the data presented in Figure 1 and Table S1 may not be correct because many times the lysis observed in the spot test is named Lysis from without (LFW), which is a lysis associated with bacterial adsorption when a high MOI is used. This is quite standard for phage researchers and the Lysis from without phenomena is well described in the literature.
- Please provide the accession number for the sequenced bacterial strains (isolates)
- Line 197: Methodology for this not very clear, see comment below (Line 489)
- Line 422: Where do the phages used in this study come from? What is the origin? This information and also the bacterial strains and culture conditions should be mentioned in the M&M section
- M&M section lacks a lot of information; some examples: How were the phages propagated? Were the phages purified? How was the phage concentration determined?
- Line 451: replace “future” with subsequent or next because the “future” here seems something that was not yet done
- Line 482: timepoint post phage-infection? Please clarify
- Lines 482-486: Repeated sentences
- Line 489: I was not able to clearly understand the assay because Figure S5 is not clear, very difficult to see the differences among plates. Better images should be provided
- Line 522: Is this adsorption assay performed in vitro or in vivo? Could you provide a reference?

Reviewer #2 (Remarks to the Author):

Ashworth et al. present a thorough translational study evaluating in a mouse model of pneumonia the efficacy of a phage cocktail against pan-resistant *Pseudomonas aeruginosa* infection. The two main findings are the emergence of phage resistance variants within the lungs in the absence of phage treatment as well as evidence for phage steering, i.e. antibiotic re-sensitization as trade-off phage resistant development.

The manuscript is well written, the experiments well designed and the conclusions of relevance when it comes to think of phage as a therapy. Other papers evaluating phage therapy in MDR strains came with the idea of phage steering (e.g. PMIDs 35537278, 33432151, 36165776)

MAJOR CONCERNS

I have the following major concerns that authors should address.

- 1) Please describe the phages within the phage cocktail in terms of taxonomy, morphology, origins, and sequence.
- 2) Please provide in vitro evidence that the phage cocktail achieve synergy compared to PELP20 alone as suggested e.g. in Fig. 2&3 lungs (time-kill curves or one-step growth curves). What could be the mechanisms to explain the observed difference of efficacy between phage PELP20 alone and the cocktail.
- 3) Re-sensitization studies: in my opinion disk diffusion assays are fine to be used as screening tools, but are not sufficient to draw conclusions as to the magnitude of re-sensitization. I propose the authors using either broth dilution methods or E-test to determine the MICs and investigate effect of phage resistance on antimicrobial susceptibilities. Fig 5 & 6 should be redrawn accordingly presenting MICs results
- 4) Authors should characterize using time kill curves the re-sensitization of the phage resistance strains to antibiotics. I would be interested to know whether re-sensitize strains are tolerant to the bactericidal effect of both bactericidal antibiotics (tolerance/persisters) or fully sensitive again to the killing.
- 5) Authors should sequence re-sensitize strains and look for genetic trade-off mechanisms? e.g. among the re-sensitize strains to meropenem whether *oprD* operon harbors mutations?
- 6) it is unclear what the authors meant in the last series of animal experiments with the concentration of antibiotics (meropenem 8ug/ml) and tobramycin (64ug/ml) (lines 274). Are those antibiotic blood levels targets to be achieved, are those concentration doses (but should be per kg) being given. In addition, this does not go along Fig legend 7. Please clarify.

MINOR

- 1) Authors should discuss the limitations of their study. e.g. the diversity of the panel used to evaluate the spectrum of the cocktail is low. Much of the strains are from keratitis.
- 2) The cocktail seem to add efficacy in treating animals with pneumonia compared to phage PELP20 alone. Can authors present their suggestion how to rationally compose a phage cocktail based on their observations.

3) Authors should update reference 4 (1999) on *P. aeruginosa* bacteremia with the following recent ones

Diekema DJ, et al. 2019. The microbiology of bloodstream

infection: 20-year trends from the SENTRY

antimicrobial surveillance program. *Antimicrob Agents Chemother* 63:e00355-19.

<https://doi.org/10.1128/AAC.00355-19>.

Alexis Tabah et al. *Intensive Care Med.* 2023 Feb;49(2):178-190. doi: 10.1007/s00134-022-06944-2. Epub 2023 Feb 10.

Epidemiology and outcomes of hospital-acquired bloodstream infections in intensive care unit patients: the EURO-BACT-2 international cohort study.

Reviewer #3 (Remarks to the Author):

Review of Ashworth et al. Phage therapy re-sensitizes *P. aeruginosa* infections to antibiotics in vivo.

In response to increasing concerns about antibiotic resistant pathogens, there has been a resurgence in interest and the application of phage for treating bacterial infections. This report can be an important contribution to our understanding of phage therapy. It has a number of virtues. The bacteria at which the phage therapy is directed, *Pseudomonas aeruginosa* is a nosocomial pathogen and a major source of morbidity and mortality of people suffering from cystic fibrosis. The phage used for the experiments have properties which them fine candidates for treating many different strains of *P. aeruginosa*, and thereby could be available for clinical use in humans. The experiments using phage to treat respiratory infections in mice are well designed. A particularly important element of this design is the treatment of established infections. Much of the failing of most experimental studies of phage (and antibiotic) therapy is that they provide treatment before the infection becomes established.

Based on the above criteria, we support the publication of this report in Nature Communications but cannot and do not recommend its publication in its current form and would like to see at least one more experiment investigating the mechanisms of at least one of the interesting observations made in this study. We would also like a more detailed explanation of several parts. Please see our below questions and suggestions.

Major Questions and Suggestions:

1. In Figure 1, the authors present the results of experiments screening the host range of four phages they use in their experiments with a large number of clinical isolates and naturally occurring *P.*

aeruginosa. Using a spot assay on soft agar lawns, they classify the response of the tested bacteria as Susceptible, Intermediate and Resistant by whether the zones of inhibition are clear, turbid, or non-existent, respectively. What is the nature of the "intermediate" states? Are only some of the phage in the lysate adsorbing to the bacteria? Are the infections aborted by some of the bacteria? Could the zones of clearance be due to bacteriocins or from lysis-from-without? Could the phage be lysogenic on those bacteria upon which they generate turbid areas? Could there be high rates of mutation to resistance? Could there be restriction-modification? More detail about the nature of the Intermediate susceptibility is essential and will likely require more experimentation.

2. *P. aeruginosa* has a number of virulence factors including toxins. For clinical use in Humans, phage lysates must be purified to remove these toxins and host cell proteins. Were the lysates used here cleaned up in this way?

3. A mechanistic answer for why the bacteria in the lung of the mock-treated mice are resistant to the phage would greatly increase the impact of this article.

4. A mechanistic investigation as to how the antibiotic-phage resistance tradeoff works would greatly increase the impact of this article.

5. The change in disc diffusion size is not a very good measure of antibiotic sensitivity. A true measure of MIC would be appreciated and much more translational than the change in disc diffusion diameter.

6. Why were late intranasal and early intravenous treatment not performed?

7. A genotypic analysis of the diversity generated by infection alone and the bacteria under phage treatment should have been performed. It looks like there is a lot of local adaptation of *P. aeruginosa* B9 occurring based on organ colonization.

Minor Questions and Suggestions:

1. *P. aeruginosa* are often lysogenic for temperate phage and infection with lytic phages can induce these temperate phages, which would be seen as plaques in the assays estimating phage densities. It would be useful to know if that is not the case here.

2. More information needs to be presented about the evolved, host range mutant, phage.

3. More information needs to be presented about the phages in the cocktail. Does resistance emerge to all three? Do the phages use different receptors?

4. What are the units in Line 130, CFU/mL?

5. The authors conclude in lines 175-178 and line 190 that PELP 20 is less effective via intranasal administration and suggest that this is due to its limited spread to other organs. What data is the conclusion being drawn from?

6. *Pseudomonas* infections tend to be heterogeneous with different phenotypic changes occurring and a wealth of genomic diversity evolving during the course of infection, the authors should address this explicitly in the discussion.

7. The authors should address why their cocktail contained the phages at such different densities.
8. The authors should consider the dynamics of individual phages within the cocktail in future studies during treatment experiments (such as Figures 2F and 3F) and not just PELP20 and the cocktail. They missed many interesting underlying dynamics due to the design of this study.

Reviewer #4 (Remarks to the Author):

The manuscript provides noteworthy results derived from an intensive experimental work that sheds light into the dynamics of phage-host interaction in vivo and supports the concept of “phage steering”. However, the research is not entirely original, since it has been previously showed that bacteriophages can resensitize bacteria to antibiotics, in vivo (eg. <https://www.nature.com/articles/s41564-020-00830-7>). Nevertheless, the article presents a significant amount of data, some of them quite original, that could be better exploited to demonstrate the in vivo dynamic interaction between phages and bacteria.

Major concerns

1. Originally of the work: Bacteria resensitization to antibiotics, following phage therapy, in vivo, has already been demonstrated and the authors should acknowledge previous work and better exploit the novelty of their work.
2. The observation that bacteria present already resistance to phages before phage therapy and that depends on the biological niche is a very relevant result for phage therapy and also original. This aspect should be better explored and explained. Is this due to the oxidative stress present in the lung environment? Are the bacteria colonizing lung tissues forming biofilms?
3. If the bacteria in the lungs are 100% resistant to the phages how do the authors explain the significant phage amplification in the lungs (line 165)?
4. The claim that phage treated bacteria alter their outer membrane proteins resulting in phage resistance should be better supported by the genome sequencing studies.
5. The genome analysis of the resistant variants isolated from the different organs of phage treated and non-treated mice should be better exploited and used not only to support phage resistance phenotype but also to explain the antibiotic susceptibility.

Minor concerns

1. The lytic spectra analysis was based on the spot test that overestimates the lytic ranges due to lysis from without phenomena that is very common in *P. aeruginosa* phages. EOP is a more accurate way to determine phage lytic ability. However, it would be tremendously time-consuming to perform EOP in the high number of bacterial isolates, so this limitation should be discussed.
2. Some experimental details are lacking, for example, how many colonies were retrieved from each assay, to assess phage susceptibility?

3. The in vivo experimental plan also lacks details. How many mice were used per group? Were the number of animals minimized with statistical relevance? What were the ethical considerations in the animal experimentation?

We also extend our thanks to the reviewers for considering our manuscript and commenting on interesting results as well as complimenting our experimental design. We appreciated their insights on complementary in vitro assays and have included these along with extensive additional data to address any concerns.

REVIEWER COMMENTS

Reviewer #1 (Remarks to the Author):

This manuscript studies the potential of phages for the treatment of pan-resistant P. aeruginosa infections using an in vivo model. The authors further study the re-sensitization to antibiotics caused by phage exposure. Although the manuscript is clear, well written and interesting results were achieved, it is long known that phages can revert antibiotic resistance due to mutations that occur in bacteria after phage exposure and so, the concept is not new. There are also some other concerns described below:

- Lines 110-119, line 413 and Table S1: The spot assay is not the appropriate method to assess the infectivity and host range of phages and can lead to false positives. The methodology here is not ok and so, the conclusions based on this assay cannot be drawn. The spot test always has to be complemented with Efficiency of plating assays. If a positive result is observed in the spot test (positive result), then serial dilutions have to be plated to see if individual phage plaques are observed in the bacterial lawn. A positive result in the spot test does not mean that the phage is able to infect and replicate in that particular strain. Phage plaques have to be observed to be able to draw that conclusion. In this case, the Authors did not perform the serial dilutions and the data presented in Figure 1 and Table S1 may not be correct because many times the lysis observed in the spot test is named Lysis from without (LFW), which is a lysis associated with bacterial adsorption when a high MOI is used. This is quite standard for phage researchers and the Lysis from without phenomena is well described in the literature.

While we agree with Reviewer 1 that efficacy of plating is the gold standard and have performed EOP assays when discussing phage resistance seen in our *in vivo* model (figure 4 and 5, line 180-207), Figure 1 and Table S1 was an initial screen to show if the phages within our cocktail show some efficacy across a wide range of *P. aeruginosa*, and conducting EOP for over 400 isolates would be a massive undertaking. We have addressed this limitation in the discussion (line 489-494) as Reviewer 4 also highlighted this issue while also noting that “it would be tremendously time-consuming to perform EOP in the high number of bacterial isolates, so this limitation should be discussed”.

-Please provide the accession number for the sequenced bacterial strains (isolates)

For the genome data accession numbers, the overall project accession is PRJEB67471. This has been added to the methods.

- Line 197: Methodology for this not very clear, see comment below (Line 489)

This has been addressed below.

- *Line 422: Where do the phages used in this study come from? What is the origin? This information and also the bacterial strains and culture conditions should be mentioned in the M&M section*

The bacterial strains included in the initial screen are listed in supplementary table 1 and culture conditions are now described on line 502-511 in the materials and methods section. The strain used in the mouse model and subsequent experiments is described on line 552. We have now included an additional table describing the phages used in the cocktail in more detail including origin, morphology, and receptor target (supplementary table 2) and this table is referenced in the main text on line 94 and in the methods section line 502 and 513.

- *M&M section lacks a lot of information; some examples: How were the phages propagated? Were the phages purified? How was the phage concentration determined?*

These details have now been added to the methods section, the phages were "Phages were propagated on strain PA01 lawns via double-layer agar method. Phages were collected by adding 10ml of PBS to the double agar plates and left to incubate for 5hr. Then the lysate was filtered using a 0.22µm filter to remove any live bacteria. Phage concentration was determined by plating serial dilutions of each phage onto bacterial PA01 lawns via double-layer agar method, where phage titres were then enumerated. Each phage PELP20, PNM, PT6, and 14/1 were diluted to the appropriate PFU/ml with PBS." line 513-538.

- *Line 451: replace "future" with subsequent or next because the "future" here seems something that was not yet done*

We have replaced the word "future" with the word "subsequent" line 555.

- *Line 482: timepoint post phage-infection? Please clarify*

We have amended the above sentence to "Each isolate harvested from mice sacrificed 48hr post-infection" line 587.

- *Lines 482-486: Repeated sentences*

We have deleted the repeated sentences.

- *Line 489: I was not able to clearly understand the assay because Figure S5 is not clear, very difficult to see the differences among plates. Better images should be provided*

We have performed EOP assays as suggested rather than using a score based on differences in lysis, making this figure no longer necessary. The EOP data is shown in Figure 4 and Figure 5.

- *Line 522: Is this adsorption assay performed in vitro or in vivo? Could you provide a reference?*

The adsorption assay was performed *in vitro* :

1. Altamirano, F.G. et al. Bacteriophage-resistant *Acinetobacter baumannii* are resensitized to antimicrobials. *Nature Microbiology* 6, 157-+ (2021).

Reviewer #2 (Remarks to the Author):

*Ashworth et al. present a thorough translational study evaluating in a mouse model of pneumonia the efficacy of a phage cocktail against pan-resistant *Pseudomonas aeruginosa* infection. The two main findings are the emergence of phage resistance variants within the lungs in the absence of phage treatment as well as evidence for phage steering, i.e. antibiotic re-sensitization as trade-off phage resistant development.*

The manuscript is well written, the experiments well designed and the conclusions of relevance when it comes to think of phage as a therapy. Other papers evaluating phage therapy in MDR strains came with the idea of phage steering (e.g. PMIDs 35537278, 33432151, 36165776)

MAJOR CONCERNS

I have the following major concerns that authors should address.

1) Please describe the phages within the phage cocktail in terms of taxonomy, morphology, origins, and sequence.

We have addressed the above comment by adding supplementary table 2, which describes the morphology origins and accession numbers. This table is referenced in the main text on line 94 and in the methods section line 502 and 513.

2) Please provide in vitro evidence that the phage cocktail achieve synergy compared to PELP20 alone as suggested e.g. in Fig. 2&3 lungs (time-kill curves or one-step growth curves). What could be the mechanisms to explain the observed difference of efficacy between phage PELP20 alone and the cocktail.

Reviewer 2 requested evidence to back up the claim that the phage cocktail achieved synergy compared to PELP20 alone. We performed a checkerboard assay with PELP20 and every phage present in the cocktail to determine percentage inhibition of bacterial growth. We found that for all three phages (14/1, PT6 and PNM), when PELP20 was present compared to when it was absent, there was a significant increase in percentage inhibition of bacterial growth (see supplementary figure 7) methods section, line 519-527. One obvious answer to the observed difference in phage efficacy is that the phage cocktail contained a higher concentration phage compared to PELP20 alone. This limitation is addressed in the discussion line 464-469.

3) Re-sensitization studies: in my opinion disk diffusion assays are fine to be used as screening tools, but are not sufficient to draw conclusions as to the magnitude of resensitization. I propose the authors using either broth dilution methods or E-test to determine the MICs and investigate effect of phage resistance on antimicrobial susceptibilities. Fig 5 & 6 should be redrawn accordingly presenting MICs results.

We agree with the reviewer that disk diffusion assays that an MIC is a more reliable method to determine the magnitude of resensitisation which is why we have included MICs (determined using E-Test) for antibiotics meropenem and tobramycin (which we used in our *in vivo* model) section starting line 239 -272 figures 6, 7 S5. However, we would also argue that disk diffusion and clinical breakpoints are used widely in a clinical setting, making an initial screen to show that resensitisation is widespread across numerous antibiotics still useful information for this study.

4) Authors should characterize using time kill curves the re-sensitization of the phage resistance strains to antibiotics. I would be interested to know whether resensitize strains are tolerant to the bactericidal effect of both bactericidal antibiotics (tolerance/persisters) or fully sensitive again to the killing.

Whilst we agree with the reviewer that this would be interesting, this is beyond the scope of this substantial study. We have focused on clinically relevant breakpoints and *in vivo* antibiotic efficacy to show that using both these measures, the bacteria *are resensitized*. These data are also now supported by the clear increase in outer membrane permeability shown in Figure 8. Additional text has been added to the discussion to acknowledge the potential role of tolerance/persistence on line 448.

5) Authors should sequence resensitize strains and look for genetic trade-off mechanisms? e.g. among the re-sensitize strains to meropenem whether oprD operon harbors mutations?

Sequencing of the *in vivo* isolates (shown in Fig 8 and S6) was conducted and mutations were reported however, the link to antibiotic resistance was unclear and could suggest that this mechanism is epigenetic see line 310-321 "Unexpectedly, delayed treated isolates, where phage resistance occurred and re-sensitisation to antibiotics was seen, only two isolates saw evidence of any SNPs, one isolate recovered from the lungs contained a frameshift variant of FC629_09380 and an isolate from the kidneys contained a missense variant in FC629_24630 (Fig 8). While this may explain phage resistance in these isolates, this does not explain the phage resistance and subsequent re-sensitisation to antibiotics in the other delayed phage treated isolates. This suggests that the mechanism of phage resistance in post-transcriptional and differs from the isolates recovered from the early phage treated experiment. It has been shown that clinical strains

of *P. aeruginosa* infected with LPS targeting phages, tend to have mutations in regulatory genes rather than LPS biosynthesis genes unlike reference strain PA01, supporting this hypothesis²⁵.”

In order to study resensitisation further, we undertook additional phenotypic studies alongside the sequencing and identified alterations in membrane permeability in the resensitised isolates.

6) it is unclear what the authors meant in the last series of animal experiments with the concentration of antibiotics (meropenem 8ug/ml) and tobramycin (64ug/ml) (lines 274). Are those antibiotic blood levels targets to be achieved, are those concentration doses (but should be per kg) being given. In addition, this does not go along Fig legend 7. Please clarify.

The concentrations of meropenem and tobramycin used in the mouse models were the concentration doses and were as listed in the figures (1mg/kg of meropenem and 5mg/kg of tobramycin) and the following concentrations listed on lines 351 and 366 have been corrected.

MINOR

1) Authors should discuss the limitations of their study. e.g. the diversity of the panel used to evaluate the spectrum of the cocktail is low. Much of the strains are from keratitis.

We have addressed the limitation mentioned above in line 492 -494 in our discussion. The initial screen was from a genetically diverse set of *Pseudomonas aeruginosa*. As most infections are acquired from the environment, population structures based on genetics are largely similar regardless of infection source. We did also test additional strains from different infections too.

2) The cocktail seem to add efficacy in treating animals with pneumonia compared to phage PELP20 alone. Can authors present their suggestion how to rationally compose a phage cocktail based on their observations.

We think the most likely explanation for the improved efficacy of the phage cocktail was due to the higher concentration of phage present in the cocktail (1×10^{12} pfu/ml) compared to (1×10^{12} pfu/ml) vs PELP20 alone (5×10^9 pfu/ml). We used 5×10^9 pfu/ml for the single phage treatment group as it is the same concentration of PELP20 present within the cocktail showing that the other phages present in the cocktail improved efficacy. We have now addressed this limitation in discussion that looking at PELP20 at the same concentration of phage as the cocktail 1×10^{12} would be interesting to compare single phage and phage cocktail therapy line 464-469.

3) Authors should update reference 4 (1999) on P aeruginosa bacteremia with the following recent ones

We have followed this reviewer's suggestion and updated the references line 706- 713.

Diekema DJ, et al. 2019. The microbiology of bloodstream infection: 20-year trends from the SENTRY antimicrobial surveillance program. Antimicrob Agents Chemother 63:e00355-19. <https://doi.org/10.1128/AAC.00355-19>.

Alexis Tabah et al. Intensive Care Med. 2023 Feb;49(2):178-190. doi: 10.1007/s00134-022-06944-2. Epub 2023 Feb 10.

Epidemiology and outcomes of hospital-acquired bloodstream infections in intensive care unit patients: the EUROACT-2 international cohort study.

Reviewer #3 (Remarks to the Author):

Review of Ashworth et al. Phage therapy re-sensitizes P. aeruginosa infections to antibiotics in vivo.

In response to increasing concerns about antibiotic resistant pathogens, there has been a resurgence in interest and the application of phage for treating bacterial infections. This report can be an important contribution to our understanding of phage therapy. It has a number of virtues. The

bacteria at which the phage therapy is directed, Pseudomonas aeruginosa is a nosocomial pathogen and a major source of morbidity and mortality of people suffering from cystic fibrosis. The phage used for the experiments have properties which them fine candidates for treating many different strains of P. aeruginosa, and thereby could be available for clinical use in humans. The experiments using phage to treat respiratory infections in mice are well designed. A particularly important element of this design is the treatment of established infections. Much of the failing of most experimental studies of phage (and antibiotic) therapy is that they provide treatment before the infection becomes established.

Based on the above criteria, we support the publication of this report in Nature Communications but cannot and do not recommend its publication in its current form and would like to see at least one more experiment investigating the mechanisms of at least one of the interesting observations made in this study. We would also like a more detailed explanation of several parts. Please see our below questions and suggestions.

Major Questions and Suggestions:

1. In Figure 1, the authors present the results of experiments screening the host range of four phages they use in their experiments with a large number of clinical isolates and naturally occurring P. aeruginosa. Using a spot assay on soft agar lawns, they classify the response of the tested bacteria as Susceptible, Intermediate and Resistant by whether the zones of inhibition are clear, turbid, or non-existent, respectively. What is the nature of the "intermediate" states? Are only some of the phage in the lysate adsorbing to the bacteria? Are the infections aborted by some of the bacteria? Could the zones of clearance be due to bacteriocins or from lysis-from-without? Could the phage be lysogenic on those bacteria upon which they generate turbid areas? Could there be high rates of mutation to resistance? Could there be restriction-modification? More detail about the nature of the Intermediate susceptibility is essential and will likely require more experimentation.

In our initial screen to determine the host range of our four phages was performed using direct spot test lysis and any plaque with opaque plaque or with resistant bacterial colonies within the plaque were determined as intermediate (See table S1). We have addressed the limitations of direct spot test in the discussion (line 489-492) and intermediate plaques could be due reduced efficacy for that particular strain on or the Lysis from without (LFW) phenomena. While it would be interesting to explore this further, it is beyond the scope of this study (and highlighted by reviewer 4 as a tremendous amount of work to undertake). The isolates used within the main study were investigated through a range of methods (inc EOP) and therefore LFW was not evident in the in vivo isolates.

2. P. aeruginosa has a number of virulence factors including toxins. For clinical use in Humans, phage lysates must be purified to remove these toxins and host cell proteins. Were the lysates used here cleaned up in this way?

The phage lysates were passed through a 0.22µl filter to remove bacteria. However due to the high MOI of phage used in our *in vivo* models, we did not perform endotoxin removal as this process reduces the phage titre. However we also recognise the importance of purifying phage lysates and consider this to be an extremely important step to move towards clinical use which is why we have addressed this questions in the discussion starting line 461-464 "While the phage cocktail achieved high levels of efficacy, a high concentration of each phage was needed and as such the phage used in this study were not purified to remove endotoxins, a process which often reduces the phage titre. Before being able to be used clinically, this cocktail would need to undergo various purification steps to be safe to administer." It is important to note however that we did not observe any adverse effects of administration of our phage lysates in mice. Phage administered mice showed no signs of discomfort or any signs associated with LPS or toxin induced fever i.e. piloerection, hunched appearance, stary coat, lethargy.

3. A mechanistic answer for why the bacteria in the lung of the mock-treated mice are resistant to

the phage would greatly increase the impact of this article.

We have addressed this question by performing evolutionary assays of the isolate used in the mouse model in media mimicking the healthy lung environment shown in a new figure (Figure 5) section beginning line 209-232. We found that environmental factors that are present in the lung such as variable oxygen availability, and the presence of mucin and polyamines cause bacteria to adapt to their environment which results in phage resistance as a by-product. This is an important finding which may help us to study, design and test effective phage cocktails for use in the lung infections.

4. A mechanistic investigation as to how the antibiotic-phage resistance tradeoff works would greatly increase the impact of this article.

We found that isolates recovered from phage treated mice were had become resistant to phages via reduced phage adsorption and had hypothesised that the phage resistant, antibiotic re-sensitised isolates had altered outer-membrane permeability resulting in re sensitisation to a wide range of antibiotics. To investigate this further not only did we WGS all of our isolates collected *in vivo*, we also conducted three different membrane permeability assays using the dyes NPN, DIsCI and propidium iodide (see figure 8) we found that compared to the input B9 isolate, re-sensitised isolates from the lungs had significantly more permeable membranes, specifically the outer membrane.

5. The change in disc diffusion size is not a very good measure of antibiotic sensitivity. A true measure of MIC would be appreciated and much more translational than the change in disc diffusion diameter.

We agree with the reviewer that disk diffusion assays that an MIC is a more reliable method to determine the magnitude of resensitisation which is why we have included MICs (determined using E-Test) for antibiotics meropenem and tobramycin (which we used in our *in vivo* model) section starting line 239 -272 figures 6, 7 S5. However, we would also argue that disk diffusion and clinical breakpoints are used widely in the clinical diagnostics setting, making an initial screen to show re-sensitisation to numerous antibiotics very useful information for this study.

6. Why were late intranasal and early intravenous treatment not performed?

We did perform delayed intranasal treatment (see supplementary figure 3). Early intravenous treatment was not performed due to lack of clinical relevance (in a clinical setting it would be unlikely to treat a pneumonia infection immediately at the time of infection with intravenous treatment of phage). This would be of little relevance to the study and by UK Home Office regulations we are bound to adhere to 3Rs principles wherever possible to limit the number of mice used.

7. A genotypic analysis of the diversity generated by infection alone and the bacteria under phage treatment should have been performed. It looks like there is a lot of local adaptation of P. aeruginosa B9 occurring based on organ colonization.

We have WGS all isolates recovered from the *in vivo* experiments see figure 8 and S6, section beginning line 277-321. We saw local adaption of *P. aeruginosa* with a 810kb duplication and a SNP in FC629_24630 in one clone isolated from the non-phage treated lung (fig 8). Additionally, we saw several SNPS that were present in the ancestor B9 infection stock at low frequency become fixed in many of the *in vivo* adapted isolates (fig S6). These are highlighted in the text in the section titled "Genetic alterations in phage resistant isolates", Figure 8, supplementary tables 3 and 4 and the discussion.

Minor Questions and Suggestions:

1. P. aeruginosa are often lysogenic for temperate phage and infection with lytic phages can induce these temperate phages, which would be seen as plaques in the assays estimating phage densities. It would be useful to Is that not the case here.

This is a really interesting point. It is slightly unclear which section of the study the reviewer is referring to, however, if this is in reference to the phages reisolated from the mice, these were

enumerated on the original input B9 isolate (as stated in the methods). If the phages were derived from induced temperate phages within the B9 genome, B9 would not show any susceptibility to the temperate phages. This was not the case and therefore we feel it is acceptable to assume that the phages isolated were from the phage therapy treatment. In future, we would like to track individual phages and have added reference to this into the discussion.

2. More information needs to be presented about the evolved, host range mutant, phage.

Due to lack of relevance and loss of evolved range samples, we have removed direct spot test data showing the evolved phages recovered from the tissue.

3. More information needs to be presented about the phages in the cocktail. Does resistance emerge to all three? Do the phages use different receptors?

We have addressed the above comment by adding supplementary table 2, which describes the morphology origins and accession numbers. This table is referenced in the main text on line 94 and in the methods section line 502 and 513.

4. What are the units in Line 130, CFU/mL?

Yes they are cfu/ml, this has been added in.

5. The authors conclude in lines 175-178 and line 190 that PELP 20 is less effective via intranasal administration and suggest that this is due to its limited spread to other organs. What data is the conclusion being drawn from?

This data is from Supplementary figure 3 where delayed administration of intranasal PELP20 showed much less efficacy than delayed intravenous administration.

The text from lines 158-162 says “To mimic treating a MDR systemic infection, phage treatment was administered 5hr post-infection, after infection had become systemic. Intranasal administration of PELP20 5hr post infection was not effective at reducing CFU loads, however, treatment with phage cocktail significantly reduced (or cleared) CFU loads in lung, liver and blood by 24 – 48hrs post-infection, compared to mock treated mice (Figure S3).”

We do not report limited spread to other organs in this section or in line 190.

6. Pseudomonas infections tend to be heterogeneous with different phenotypic changes occurring and a wealth of genomic diversity evolving during the course of infection, the authors should address this explicitly in the discussion.

We have not discussed this heterogeneous nature of chronic *P. aeruginosa* lung infections in this study as this is not a model that mimics chronic infection. We have added text to the discussion to acknowledge this “Furthermore, *P. aeruginosa* displays both genetic and phenotypic heterogeneity, thereby potentially making chronic lung infections a greater phage treatment challenge.” (line 434-436).

7. The authors should address why their cocktail contained the phages at such different densities.

We chose phage densities as described in methods section 533-538, Figure S7. This was due to in vitro efficacy however we acknowledge calculating MOI is a challenge after 5hr of infection progression in vivo.

8. The authors should consider the dynamics of individual phages within the cocktail in future studies during treatment experiments (such as Figures 2F and 3F) and not just PELP20 and the cocktail. They missed many interesting underlying dynamics due to the design of this study.

We have addressed this limitation in the discussion line 464 - 469 “Additionally, while the phage cocktail was found to be more efficacious than single phage treatment, the overall concentration of phages within the cocktail was higher (1×10^{12} pfu/ml) vs PELP20 alone (5×10^9 pfu/ml). While this provides evidence that the other phages improve efficacy, it would be interesting to see if administering PELP20 at 1×10^{12} would achieve the same efficacy as the phage cocktail, to further explore phage-phage interactions.”

Reviewer #4 (Remarks to the Author):

The manuscript provides noteworthy results derived from an intensive experimental work that sheds light into the dynamics of phage-host interaction in vivo and supports the concept of “phage steering”. However, the research is not entirely original, since it has been previously showed that bacteriophages can resensitize bacteria to antibiotics, in vivo (eg. <https://www.nature.com/articles/s41564-020-00830-7>). Nevertheless, the article presents a significant amount of data, some of them quite original, that could be better exploited to demonstrate the in vivo dynamic interaction between phages and bacteria.

Major concerns

1. Originally of the work: Bacteria resensitization to antibiotics, following phage therapy, in vivo, has already been demonstrated and the authors should acknowledge previous work and better exploit the novelty of their work.

We had highlighted previous work (including the above mentioned reference in line 77 and 78 as well as including the following references:

- Gurney, J. et al. Phage steering of antibiotic-resistance evolution in the bacterial pathogen, *Pseudomonas aeruginosa*. *Evolution Medicine and Public Health*, 148-157 (2020).
- Chan, B.K. et al. Phage selection restores antibiotic sensitivity in MDR *Pseudomonas aeruginosa*. *Scientific Reports* 6, 8 (2016).

We have though altered the manuscript title to better highlight the novelty of the work.

2. The observation that bacteria present already resistance to phages before phage therapy and that depends on the biological niche is a very relevant result for phage therapy and also original. This aspect should be better explored and explained. Is this due to the oxidative stress present in the lung environment? Are the bacteria colonizing lung tissues forming biofilms?

We have addressed this question by performing evolutionary assays of the isolate used in the mouse model in media mimicking the healthy lung environment shown in a new figure (Figure 5) section beginning line 209 - 232. We found that environmental factors that are present in the lung such as variable oxygen availability, and the presence of mucin and polyamines cause bacteria to alter to adapt to their environment which results in phage resistance as a by-product.

3. If the bacteria in the lungs are 100% resistant to the phages how do the authors explain the significant phage amplification in the lungs (line 165)?

The bacteria in the lungs are resistant to the input phages at 48hr post infection. We administered phage either at 0hr or 5hr post infection when the bacteria would still be sensitive to the phage, allowing the phage to replicate resulting in the amplification seen.

4. The claim that phage treated bacteria alter their outer membrane proteins resulting in phage resistance should be better supported by the genome sequencing studies.

We have supported this claim by WGS all of our isolates collected *in vivo*, we also conducted three different membrane permeability assays using the dyes NPN, DIsCI and propidium iodide (see figure 8) we found that compared to the input B9 isolate, re-sensitised isolates from the lungs were had significantly more permeable membrane, specifically the outer membrane.

5. The genome analysis of the resistant variants isolated from the different organs of phage treated and non-treated mice should be better exploited and used not only to support phage resistance phenotype but also to explain the antibiotic susceptibility.

We found that isolates recovered from phage treated mice had become resistant to phages via reduced phage adsorption and we hypothesised that the phage resistant, antibiotic re-sensitised isolates had altered outer-membrane permeability resulting in re-sensitisation to a wide range of antibiotics. To investigate this further not only did we WGS all of our isolates collected *in vivo*, we also conducted three different membrane permeability assays using the dyes NPN, DIsCI and propidium iodide (see figure 8). We found that compared to the input B9 isolate, re-sensitised isolates from the lungs had significantly more permeable membranes, specifically the outer membrane.

Minor concerns

1. The lytic spectra analysis was based on the spot test that overestimates the lytic ranges due to lysis from without phenomena that is very common in P aeruginosa phages. EOP is a more accurate way to determine phage lytic ability. However, it would be tremendously time-consuming to perform EOP in the high number of bacterial isolates, so this limitation should be discussed.

We agree with the reviewer and performed EOP on all isolates recovered from the in vivo model however as the reviewer points out performing EOP on over 400 isolates would be time consuming so we added to the discussion on line 489 - 494)

2. Some experimental details are lacking, for example, how many colonies were retrieved from each assay, to assess phage susceptibility?

We have addressed this comment in the methods section line 574 "At 48hr, bacterial isolates (three per treatment group per tissue site) were harvested from any organs where bacteria could be cultured."

3. The in vivo experimental plan also lacks details. How many mice were used per group? Were the number of animals minimized with statistical relevance? What were the ethical considerations in the animal experimentation?

For each *in vivo* experiment, 10 mice per group per timepoint were used, in two independent experiments unless stated otherwise. The methods section has been amended to address this comment line 539-560. All mouse work was approved and performed in accordance with the regulations of the Home Office Scientific Procedures Act (1986), project licence P86De83DA and the University of Liverpool Ethical and Animal Welfare Committee.

REVIEWER COMMENTS

Reviewer #1 (Remarks to the Author):

All the comments have been addressed in the revised version.

Some remarks:

- The title of table S1 should be changed to Spot test assay against MDR *P. aeruginosa* strains or similar because the current one "Susceptibility of *P. aeruginosa* isolates to phages" is not correct as the test carried out by the authors do not allow to directly draw conclusions on phage susceptibility without serial dilutions.

- The title Phage sensitivity screening in Methods section is also not correct and the authors mention plaque formation but as no dilutions were made, no phage plaques are observed. This should be corrected accordingly and changed to inhibition zone instead of plaques. This inaccuracy related to the fact that the authors did not perform EOP is propagated throughout the manuscript

Reviewer #2 (Remarks to the Author):

I would like to congratulate the authors for the additional work performed and for answering Reviewers' concerns.

I have at this stage only one additional comment:

Did resensitization also occur among the isolates evolved in vitro in media mimicking the lung niche (Figure 5)? Were antibiotic disk diffusion assay /MIC determinations also performed on phage resistant variants that emerged in those media (e.g. polyamines (spermidine-200ng/ml, spermine-32.5ug/l, and putrescine-616ug/l) or mucin (1.2mg/ml))?

Reviewer #3 (Remarks to the Author):

We feel that the authors have adequately addressed our general and specific comments and based on this criteria, now feel the manuscript is ready for publication.

Reviewer #4 (Remarks to the Author):

The authors have addressed quite well the question raised. I have no further remarks

Response to Reviewers Comments.

Once again we wish to thank the reviewers for their time and effort on this manuscript. We are delighted that all 4 reviewers agree that we have fully addressed the previous review comments. A very small number of additional comments were made by reviewers 1 and 2. We are delighted to fully address these.

In addition to the changes below, we have also made a small change to the abstract and the subheadings to conform to editorial policies.

Reviewer #1 (Remarks to the Author):

All the comments have been addressed in the revised version.

Some remarks:

- The title of table S1 should be changed to Spot test assay against MDR *P. aeruginosa* strains or similar because the current one "Susceptibility of *P. aeruginosa* isolates to phages" is not correct as the test carried out by the authors do not allow to directly draw conclusions on phage susceptibility without serial dilutions.

The title has been altered to "Supplementary Table 1) Direct spot test assay of *P. aeruginosa* MDR isolates using bacteriophages PELP20, PT6, PNM, 14/1."

- The title Phage sensitivity screening in Methods section is also not correct and the authors mention plaque formation but as no dilutions were made, no phage plaques are observed. This should be corrected accordingly and changed to inhibition zone instead of plaques. This inaccuracy related to the fact that the authors did not perform EOP is propagated throughout the manuscript

The title has been altered to "Direct spot test assay". The Methods text has been altered to state:

"The screen for phage activity was determined via direct spot test, 10ul of each phage were spotted on LB plates with an overlay of each clinical isolate, and left overnight at 37 °C. The following day, the phage activity against each isolate was recorded as sensitive (+) donating clear zone of inhibition, (±) indicating turbid or fainter zone of inhibition and sensitive (-) to indicate resistance (Supplementary Table 1)."

Reviewer #2 (Remarks to the Author):

I would like to congratulate the authors for the additional work performed and for answering Reviewers' concerns.

I have at this stage only one additional comment:

Did resensitization also occur among the isolates evolved in vitro in media mimicking the lung niche (Figure 5)? Were antibiotic disk diffusion assay /MIC determinations also performed on phage resistant variants that emerged in those media (e.g. polyamines (spermidine-200ng/ml, spermine-32.5ug/l, and putrescine-616ug/l) or mucin (1.2mg/ml))?

Resensitisation of bacteria after exposure to lung components was not seen during this experiment. We consider this consistent with our hypothesis that phage drive the resensitisation as there were no phages present in this adaptation experiment. We have stated this clearly in the methods section and added a new supplementary figure (8):

“Isolates recovered from the LB broth, healthy lung media, LB supplemented with polyamines and LB broth supplemented with mucin were also tested for antibiotic sensitivity via E-TEST for meropenem and tobramycin, with little to no re-sensitisation seen (Supplementary Fig. 8).”